# Antiferroelectric negative capacitance from a structural phase transition in zirconia

Michael Hoffmann[1,12,13 ✉], Zheng Wang[2,13], Nujhat Tasneem[2,13], Ahmad Zubair[3],
Prasanna Venkatesan Ravindran[2], Mengkun Tian[4], Anthony Arthur Gaskell[2], Dina Triyoso[5],
Steven Consiglio[5], Kandabara Tapily[5], Robert Clark[5], Jae Hur[2], Sai Surya Kiran Pentapati[2], Sung Kyu Lim[2],
Milan Dopita[6], Shimeng Yu[2], Winston Chern[3,7], Josh Kacher[8], Sebastian E. Reyes-Lillo[9], Dimitri Antoniadis[3],
Jayakanth Ravichandran[10], Stefan Slesazeck[1], Thomas Mikolajick[1,11] & Asif Islam Khan[2,8 ✉]

Crystalline materials with broken inversion symmetry can exhibit a spontaneous electric polarization, which originates from a microscopic electric dipole moment. Long-range polar or anti-polar order of such permanent dipoles gives rise to ferroelectricity or antiferroelectricity, respectively. However, the recently discovered antiferroelectrics of fluorite structure ($HfO_2$ and $ZrO_2$) are different: A non-polar phase transforms into a polar phase by spontaneous inversion symmetry breaking upon the application of an electric field. Here, we show that this structural transition in antiferroelectric $ZrO_2$ gives rise to a negative capacitance, which is promising for overcoming the fundamental limits of energy efficiency in electronics. Our findings provide insight into the thermodynamically forbidden region of the antiferroelectric transition in $ZrO_2$ and extend the concept of negative capacitance beyond ferroelectricity. This shows that negative capacitance is a more general phenomenon than previously thought and can be expected in a much broader range of materials exhibiting structural phase transitions.

[1] NaMLab gGmbH, 01187 Dresden, Germany. [2] School of Electrical and Computer Engineering, Georgia Institute of Technology, Atlanta, GA 30332, USA. [3] Department of Electrical Engineering and Computer Science, Massachusetts Institute of Technology, Cambridge, MA 02142, USA. [4] Institute for Electronics and Nanotechnology, Georgia Institute of Technology, Atlanta, GA 30332, USA. [5] TEL Technology Center, America, LLC, 255 Fuller Rd., Suite 214, Albany, NY 12203, USA. [6] Department of Condensed Matter Physics, Faculty of Mathematics and Physics, Charles University, Ke Karlovu 5, 12116 Prague, Czech Republic. [7] Izentis LLC, PO Box 397002 Cambridge, MA 02139, USA. [8] School of Materials Science and Engineering, Georgia Institute of Technology, Atlanta, GA 30332, USA. [9] Departamento de Ciencias Físicas, Universidad Andres Bello, Santiago 837-0136, Chile. [10] Department of Chemical Engineering and Materials Science, University of Southern California, Los Angeles, CA 90089, USA. [11] Institute of Semiconductors and Microsystems, TU Dresden, Dresden, Germany. [12] Present address: Department of Electrical Engineering and Computer Sciences, University of California, Berkeley, CA 94720, USA. [13] These authors contributed equally: Michael Hoffmann, Zheng Wang, Nujhat Tasneem. ✉email: hoffmann@berkeley.edu; asif.khan@ece.gatech.edu

Antiferroelectric materials are promising for diverse applications ranging from energy harvesting[1] and solid state cooling devices[2] over electromechanical transducers[3] to energy storage supercapacitors[4,5]. First predicted in 1951[6], antiferroelectricity was subsequently discovered in the archetypal perovskite oxide, lead zirconate ($PbZrO_3$)[7,8]. Ever since, the range of materials has been expanded to two-dimensional hybrid perovskites[9], interfacially engineered heterostructures and superlattices[10], fluorite structure binary oxides[11,12], and more have been predicted by first principles calculations[13,14]. However, compared to their ferroelectric counterparts, antiferroelectrics have remained less explored and understood so far, despite their intriguing properties and rich phase transition phenomena[14-16]. Therefore, antiferroelectric materials hold a large untapped potential for the discovery of emergent phases for example in antiferroelectric oxide heterostructures, which have been investigated in this work.

Antiferroelectricity is characterized by a distinctive, double hysteresis loop in the macroscopic electric polarization $P$ vs. electric field $E_a$ characteristics as shown in Fig. 1a based on a Kittel-model (see Supplementary Methods)[6]. From a thermodynamic perspective, such a macroscopic response can be described by a free energy ($G$)-polarization landscape as shown in Fig. 1b for $E_a = 0$, where the only stable state is the macroscopically non-polar ground state (A) at $P = 0$. However, when an electric field $E_a = E_1$ is applied, see Fig. 1c, an energy barrier emerges between the non-polar state M and the polar state N. This forbidden region of negative free energy curvature ($d^2G/dP^2 < 0$) between points B and C is thermodynamically unstable[6,15]. Note that the relative energies of the non-polar and polar states in Fig. 1c, d are dependent upon the antiferroelectric material and also the magnitude of the electric field, i.e. the polar phase will become lower in energy than the non-polar phase for even higher applied fields. According to theory, the non-linear permittivity and thus capacitance of a material is proportional to $(d^2G/dP^2)^{-1}$, which means that at the antiferroelectric transition the capacitance of the material would become negative if stabilized in a larger system[17,18]. Note that we always mean "negative *differential* capacitance" when we write "negative capacitance" in this context. Due to the inversion symmetry of the non-polar ground state, the same transition occurs when an opposite electric field $E_a = -E_1$ is applied, see Fig. 1d. Therefore, two separate and symmetric regions of negative capacitance can be predicted at the antiferroelectric non-polar to polar phase transition as shown by the dotted lines in Fig. 1a where $dP/dE_a < 0$.

Similar 'S'-shaped $P$–$E$ curves have been derived from the Landau-Ginzburg-Devonshire theory of ferroelectric phase transitions many decades ago but were previously understood to be inaccessible to experiments due to their unstable nature[19]. Only recently, it was suggested that one could access these forbidden thermodynamic regions in ferroelectric/dielectric heterostructures using pulsed electrical measurements[20,21]. In such a structure, the positive capacitance of the dielectric layer can stabilize the region of negative capacitance and prevent the screening of bound polarization charge by free electrons in the metal electrodes. Here, we apply a similar approach to first probe the inaccessible region of a non-polar to polar structural phase transition in an antiferroelectric oxide. While experimental insights into such thermodynamic instabilities are of fundamental interest to materials science, they are also useful for prospective applications since the resulting negative capacitance can be used to amplify voltage signals in electronic devices and circuits[18,22].

Historically, antiferroelectricity has been related to the antipolar alignment of microscopic electric dipoles in the unit cell[6]. For example, in the ground state of antiferroelectric $PbZrO_3$, two adjacent columns of Pb ions point in the same direction, while the

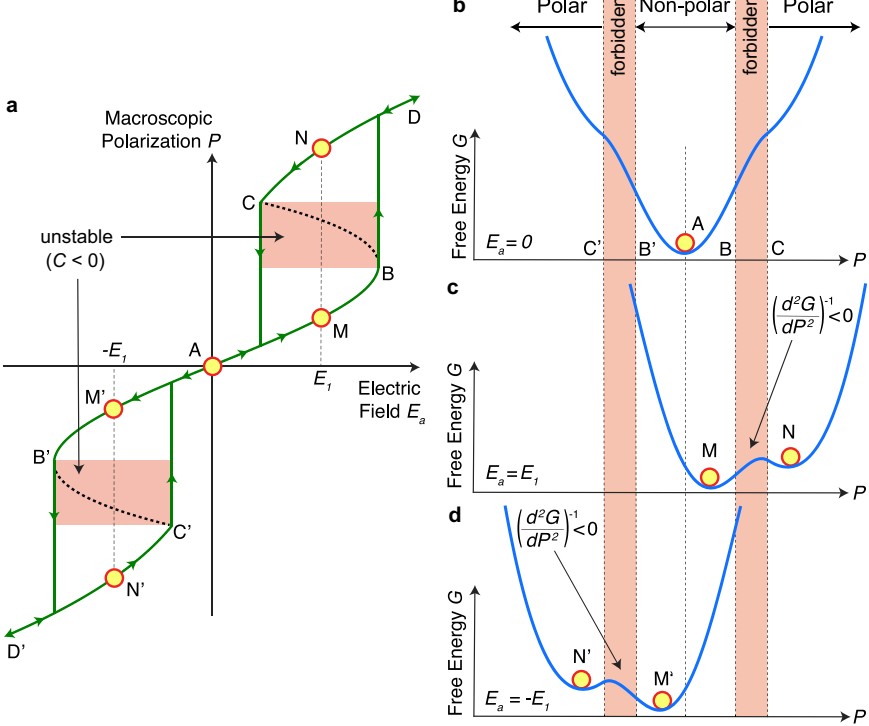

**Fig. 1 Origin of antiferroelectric negative capacitance. a** The polarization $P$-electric field $E_a$ characteristics of an antiferroelectric material. The segment BAB' corresponds to the non-polar, antiferroelectric ground state, and segments CD and C'D' correspond to the polar phase. Segments BC and B'C' represent the unstable negative capacitance (C < 0) regions. At $Ea = E_1$, the antiferroelectric has two stable states: M and N. **b–d** The antiferroelectric free energy landscape at $E_a = 0$ (**b**), $E_1$ (**c**) and $-E_1$ (**d**). $d^2G/dP^2 < 0$ in the $P$-range corresponding to BC and B'C' 'forbidden' regions.

next two columns have antiparallel alignment[7]. The application of a sufficiently large electric field then leads to a spontaneous alignment of the anti-parallel dipoles resulting in a transition to a polar ferroelectric state[23]. In a similar way, first principles calculations of ferroelectric/paraelectric heterostructures have predicted antiferroelectric-like negative capacitance regions due to the transition between a striped domain pattern and a ferroelectric monodomain state[24]. In contrast, the newly discovered $HfO_2$ and $ZrO_2$ based antiferroelectrics of fluorite structure transcend the classical definition of antiferroelectricity[11,12], since their ground state does not exhibit anti-polar order but is microscopically non-polar[25]. Therefore, the original Kittel-model in Fig. 1a might not give a precise microscopic picture of antiferroelectricity in fluorite structure oxides and thus cannot explain the results quantitatively. However, the qualitative prediction of a negative capacitance from the non-polar to polar phase transition still applies. Note that this is fundamentally different from previous investigations in ferroelectrics, where no phase transition or change of symmetry occurs in the forbidden region of the energy landscape[21].

Here, we investigate this unique type of non-polar to polar antiferroelectric transition in $ZrO_2$ as a model system, which is of significant technological importance due to its compatibility with semiconductor manufacturing and thickness scalability to the nanometer regime[12]. Additionally, no doping of $ZrO_2$ is needed to induce antiferroelectricity in contrast to $HfO_2$ based thin films[11,26]. At room temperature, the stable bulk phase of $ZrO_2$ is the non-polar monoclinic $P2_1/c$ phase, which can be suppressed in thin films of around 10 nm thickness and below, which favor the non-polar tetragonal $P4_2/nmc$ phase[12]. The current understanding of the origin of antiferroelectricity in $ZrO_2$ is that the non-polar tetragonal $P4_2/nmc$ phase undergoes a first-order structural phase transition into the polar orthorhombic $Pca2_1$ phase by application of an electric field of around 2-3 MV cm$^{-1}$ [25,26]. The polar orthorhombic $Pca2_1$ phase has been shown to be responsible for the ferroelectric behavior observed in $HfO_2$ based thin films[27]. While obtaining definitive experimental proof of such a field-induced first-order phase transition has proved difficult so far[28], this mechanism is consistent with first principles calculations[25,29] as well as composition- and temperature-dependent experimental results[26]. While the microscopic switching pathway between the $P4_2/nmc$ and $Pca2_1$ phases is still unclear, it has been suggested to include an intermediate phase of orthorhombic $Pmn2_1$ symmetry for $Hf_{0.5}Zr_{0.5}O_2$[30].

## Results and discussion

### Characterization of antiferroelectricity in $ZrO_2$.
For basic antiferroelectric characterization, $TiN/ZrO_2/TiN$ capacitors were fabricated as described in methods. Grazing incidence X-ray diffraction (GIXRD) measurement results presented in Supplementary Fig. 1 confirm that the crystalline $ZrO_2$ layer is in the non-polar tetragonal $P4_2/nmc$ phase in the as fabricated capacitors. Figure 2a and Supplementary Fig. 2a show the antiferroelectric double hysteresis loops in the polarization-electric field characteristics for a 10 nm and 5 nm $ZrO_2$ layer, respectively, measured using a ferroelectric tester. Since the structural data suggests that our $ZrO_2$ films are fully tetragonal without applied voltage, the measured double hysteresis loops can only be explained by a field-induced structural transformation into a polar ferroelectric phase. While we cannot directly determine the symmetry of this polar phase at high electric field, there is substantial evidence in literature which suggests that it is of orthorhombic $Pca2_1$ symmetry[26]. For example, a temperature-dependent phase transition from the non-polar $P4_2/nmc$ phase to the polar $Pca2_1$ phase in similar $ZrO_2$ thin films has been

experimentally observed with in situ high-temperature X-ray diffraction[31]. Furthermore, $ZrO_2$ can be stabilized in the $Pca2_1$ phase even at room temperature under certain processing conditions[32]. Lastly, the field-induced transition from the $P4_2/nmc$ phase to the $Pca2_1$ phase has been directly observed in $Hf_{0.5}Zr_{0.5}O_2$ thin films[33]. From these experimental data and previous first principles calculations[25,29], it seems reasonable to conclude that the polar phase observed at high electric field in Fig. 2a is of $Pca2_1$ symmetry. Furthermore, using high-resolution transmission electron microscopy (HRTEM) with in situ voltage biasing, it was directly shown that antiferroelectric $ZrO_2$ always returns to its initial non-polar $P4_2/nmc$ structure after the applied voltage is removed[28]. Since the $ZrO_2$ film investigated in ref. [28] was fabricated in the exact same way as the ones shown here, it is reasonable to assume that they also return to the initial non-polar $P4_2/nmc$ phase after each field-induced phase transition.

### Antiferroelectric/dielectric heterostructures.
However, as expected for a single layer antiferroelectric capacitor in Fig. 2a, the 'forbidden' regions (see Fig. 1) of the structural transition to the polar phase cannot be observed electrically due to their unstable nature. To access these unstable regions, a positive series capacitance can be used in the form of a dielectric layer in contact with the antiferroelectric[18,20,21]. The dielectric layer has two important functions: It prevents the injection of compensating charge which could screen the bound polarization $P$ and it creates a depolarization field in the antiferroelectric which is antiparallel to $P$, such that $E_a$ can decrease while $P$ is increasing. When the antiferroelectric enters the unstable region in such a heterostructure, the resulting negative capacitance would then increase the total capacitance of the dielectric/antiferroelectric stack beyond than that of the dielectric layer alone. Such a capacitance enhancement would thus be a clear signature of the structural transition itself. To test this prediction, we fabricated heterostructure capacitors consisting of the same $ZrO_2$ layer and a stack of $Al_2O_3/HfO_2$ dielectrics with TiN used as top and bottom electrodes using atomic layer deposition as described in methods. The intermediate $Al_2O_3$ layer was introduced to prevent the crystallization of the top $HfO_2$ layer due to templating effects from the underlying crystallized $ZrO_2$. Scanning transmission electron microscopy (STEM) analysis of a representative $TiN/ZrO_2/Al_2O_3/HfO_2/TiN$ heterostructure in Fig. 2b confirms that the $ZrO_2$ layer was stabilized in its non-polar tetragonal phase while the $HfO_2$ layer remained amorphous. The combined results of our GIXRD measurements together with STEM and nanobeam electron diffraction data (see Supplementary Fig. 3) present strong evidence for a fully non-polar tetragonal ground state of the $ZrO_2$ layer. As mentioned before, previous in situ HRTEM experiments showed that these $ZrO_2$ layers always return to their initial non-polar ground state after the applied voltage is removed[28].

### Pulsed electrical characterization.
To probe the capacitance of the heterostructure under high applied fields, we adopt a pulsed capacitance measurement technique[20,34]. We apply microsecond voltage pulses and the charge supplied by the voltage source is directly measured by integrating the measured current. Details are described in methods. Note that standard small-signal capacitance-voltage measurements are too slow at the high applied voltages needed, leading to significant charge injection or dielectric breakdown. In contrast, pulsed capacitance measurements can mitigate this, if the pulse duration is shorter (~1 μs in our experiments) than the time scale for charge injection, breakdown and related mechanisms[20].

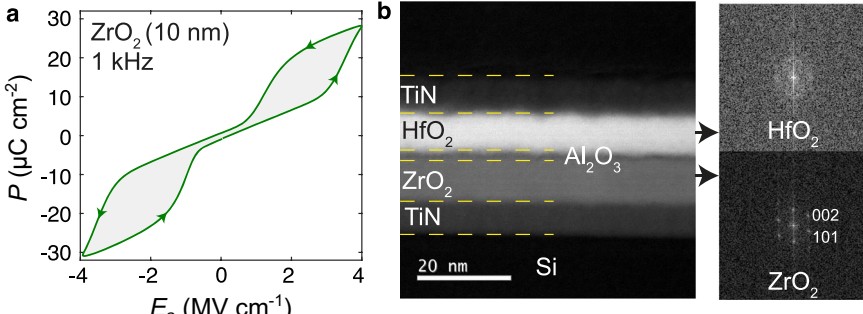

**Fig. 2 Standard characterization of antiferroelectric $ZrO_2$ thin film and heterostructure. a** Polarization $P$ vs. electric field $E_a$ characteristics of a TiN/$ZrO_2$(10 nm)/TiN capacitor measured using a standard ferroelectric tester at 1 kHz. **b** Low magnification high angle annular dark field (HAADF) scanning transmission electron microscopy (STEM) image of the cross-section of a representative TiN/$HfO_2$/$Al_2O_3$/$ZrO_2$/TiN heterostructure grown on Si showing all the layers distinguishable with clear interfaces. Fast Fourier transforms of high magnification HAADF-STEM images of the same sample show amorphous rings in the $HfO_2$ layer and discrete diffraction spots in the $ZrO_2$ layer consistent with the tetragonal <010> zone axis.

Figure 3a shows the experimental setup in which voltage pulses $V_{in}(t)$ ($t \equiv$ time) of $T = 1.1$ μs duration, different amplitudes $V_a$ were applied to the capacitors in series to an external resistor $R$. Waveforms of $V_{in}$, measured current $I$, and integrated charge ($=\int I dt$) for different amplitudes $V_a$ for a $HfO_2$(8 nm)/$Al_2O_3$(~1 nm)/$ZrO_2$(10 nm) capacitor are shown in Fig. 3b. For a given $V_a$, $Q_{max}$ is the total charge supplied by the voltage source (i.e., $Q_{max} = \int_0^T I(t)dt$). $Q_{res}$ is the residual charge when $V_{in}$ goes to zero (i.e., $Q_{res} = \int_0^\infty I(t)dt$), and accounts for charge injection and leakage. The difference between $Q_{max}$ and $Q_{res}$ is defined as $\Delta Q$, which is the actual amount of charge that is reversibly delivered to and discharged from the capacitor and determines the differential capacitance of the heterostructure as $C_{DE-AFE} = \Delta Q / \Delta V_a$. Figure 3c plots $Q_{max}$, $Q_{res}$ and $\Delta Q$ (calculated from Fig. 3b: panel 3) as functions of $V_a$. We note in Fig. 3c for positive $V_a$ that $Q_{res}$ is zero, and $\Delta Q = Q_{max}$ as expected in the absence of charge injection and leakage for a purely capacitive load. The slope of the $\Delta Q - V_a$ curve for $V_a \geq 10$ V (YZ segment) is larger than the capacitance of the constituent dielectric stack $C_{DE}$ which is shown as a slope in Fig. 3c and was measured on a separate $HfO_2$(8 nm)/$Al_2O_3$(~1 nm) capacitor fabricated under the same processing conditions (see Supplementary Fig. 4). For large negative values of $V_a$, $Q_{res}$ has a non-zero value; however, accounting for the corresponding charge injection and leakage, a similar capacitance enhancement is observed for $V_a < -9.2$ V (Y'Z' segment).

From the inversion symmetry of the measured $\Delta Q - V_a$ curves in Fig. 3c despite the asymmetric $Q_{res} - V_a$ behavior one can conclude that the capacitive behavior is unaffected by leakage currents or charge injection. Furthermore, a symmetric $\Delta Q - V_a$ curve is expected only for an ideal antiferroelectric/dielectric stack with negligible trapped charges at the interface between both layers. This contrasts with recent reports of similar ferroelectric/dielectric capacitors, where large negative trapped charge densities (comparable to the spontaneous polarization) were found at the interface[20,21]. This indicates that the initial ferroelectric polarization during fabrication in such stacks causes the trapping of charges at the interface, which is not the case for the antiferroelectric heterostructure due to the non-polar antiferroelectric ground state. Therefore, having a non-polar ground state seems beneficial for avoiding trapped interface charges, which can be detrimental when building negative capacitance devices.

**Extracting negative capacitance and energy landscape.** For a given $V_a$, the electric field in the antiferroelectric layer $E_a$ can be calculated as $E_a = (V_a - V_{DE})/t_{AFE}$, where $t_{AFE}$ is the $ZrO_2$ thickness and $V_{DE} = \Delta Q(V_a)/C_{DE}$ (see Methods for details). The

corresponding $P - E_a$ curve ($P = \Delta Q/A$ with $A$ being the area of the capacitor) of the $ZrO_2$ layer in Fig. 3d shows two separate regions of negative slope (i.e. negative capacitance) which correspond to the capacitance enhancement regions in the $\Delta Q - V_a$ curve in Fig. 3c. For comparison, the $P - E_a$ characteristics of an equivalent, single-layer $ZrO_2$ capacitor measured using a ferroelectric tester is also plotted in Fig. 3d. The locations of the negative capacitance regions in Fig. 3d coincide with the two hysteresis regions of the polarization-electric field characteristics of the stand-alone $ZrO_2$ layer, in agreement with the theoretical prediction (Fig. 1a). However, the second positive capacitance branch expected at higher fields is often not observed in experiments, since hard dielectric breakdown of the dielectric layers occurs at these high electric fields. Nevertheless, we were able to observe the second positive capacitance branch in some samples that showed a slightly higher than average breakdown field strength as can be seen in the Supplementary Fig. 9. Furthermore, in Supplementary Fig. 12 we incrementally changed the voltage pulse amplitude from 0 V → 11 V → −11 V → 0 V to investigate the reversibility of the $P - E_a$ curve. An asymmetric hysteresis emerges which seems to correlate with the observation of significant $Q_{res}$ for negative voltages in Fig. 3c. This suggests that the hysteresis is not related to the non-polar to polar phase transition itself, but that it is caused by leakage and subsequent charge trapping of a fraction of $Q_{res}$ at the antiferroelectric/dielectric interface, which leads to a shift of the apparent $P - E_a$ curve. For positive voltages, where $Q_{res}$ is low, negative capacitance is observed in both forward and backwards sweep directions in Supplementary Fig. 12.

By integrating the $P - E_a$ curve, one obtains the antiferroelectric energy landscape ($G(P) = \int E_a dP$) which is shown in Fig. 3e, where the normally forbidden regions ($G'' < 0$) of thermodynamic instability at the non-polar to polar structural phase transition can be accessed in the antiferroelectric/dielectric heterostructure. It is interesting to note here, that previous first principles calculations for the antiferroelectric transition in $ZrO_2$ found that $G''$ is always positive along the direct switching pathway between the $P4_2/nmc$ and $Pca2_1$ phase, corresponding to a cusp in the energy landscape[25]. This is in contrast to our experimental findings, where negative $G''$ regions are clearly observed (see Fig. 3e). However, recent first principles calculations have shown that there could be other switching pathways between the $P4_2/nmc$ and $Pca2_1$ phase, which can avoid the cusp in the energy landscape by traversing through an intermediate orthorhombic $Pmn2_1$ phase[30]. Therefore, our findings provide the first indirect experimental evidence that such an alternative switching pathway might exist in antiferroelectric $ZrO_2$.

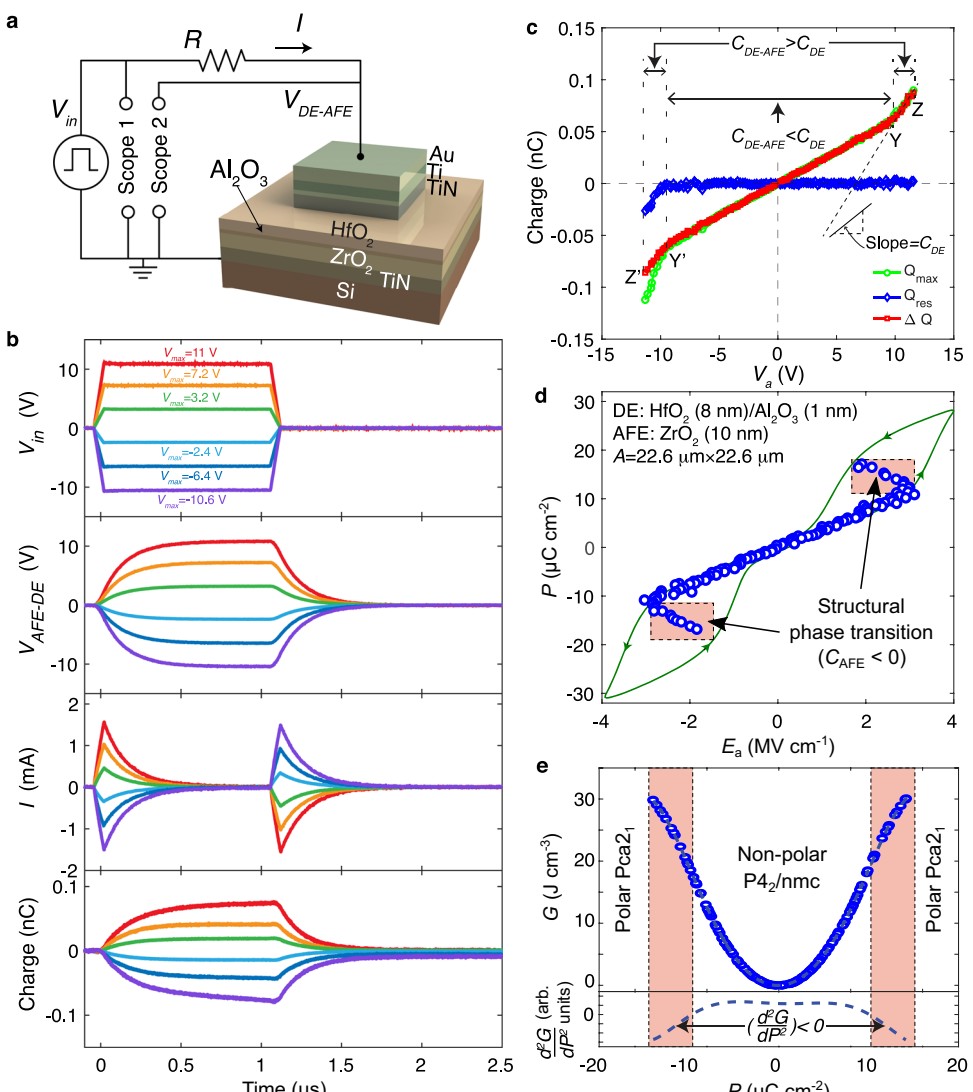

**Fig. 3 Demonstration of antiferroelectric negative capacitance. a** Experimental setup of the pulsed charge-voltage measurements on the dielectric-antiferroelectric heterostructure. $V_{in}$, $V_{DE-AFE}$, $R$ and $I$ are the applied voltage pulse, the voltage across the DE-AFE capacitor, the series resistor (5.6 kΩ) and the current through $R$, respectively. The waveforms of $V_{in}$ and $V_{DE-AFE}$ were measured using an oscilloscope at different amplitudes of the $V_{in}$ pulse. **b** Transient waveforms of $V_{in}$, $V_{DE-AFE}$, $I$ and integrated charge for a HfO$_2$(8 nm)/Al$_2$O$_3$(~ 1 nm)/ZrO$_2$(10 nm) capacitor. **c** Maximum charge $Q_{max}$, residual charge $Q_{res}$, and reversibly stored charge $\Delta Q$ as functions of maximum voltage across the DE-AFE capacitor $V_a$ measured from the waveforms shown in **b**. **d** Polarization $P$ as a function of extracted electric field $E_a$ across the ZrO$_2$ layer in a HfO$_2$(8 nm)/Al$_2$O$_3$(~ 1 nm)/ZrO$_2$ (10 nm) heterostructure capacitor. The $P$-$E_a$ characteristics of an equivalent stand-alone ZrO$_2$ capacitor measured on a conventional ferroelectric tester is also shown for comparison in the background. The negative capacitance regions ($C_{AFE} < 0$) in the $P$-$E_a$ curve correspond to the capacitance enhancement regions in the $\Delta Q$-$V_a$ curve shown in **c**. **e** Extracted energy landscape of ZrO$_2$. Second derivative of the free energy $G$ with respect to $P$ based on a polynomial fit is shown below.

**Effect of dielectric layer thickness.** Next, we changed the HfO$_2$ thickness to 5 nm, 6 nm and 10 nm while keeping Al$_2$O$_3$ and ZrO$_2$ layer thicknesses constant (~1 nm and 10 nm, respectively). For all different HfO$_2$ thicknesses, the extracted $P-E_a$ characteristics of the ZrO$_2$ layer have the same quantitative shape as shown in Supplementary Fig. 5-7. We further observed in Fig. 4a that the capacitance enhancement ($r = C_{DE-AFE}/C_{DE} > 1$) in these samples obeys the ideal capacitance matching equation $r = |C_{AFE}|/(|C_{AFE}| - C_{DE})$ with a best fit antiferroelectric negative capacitance $C_{AFE}° = -4.75$ μF cm$^{-2}$, extracted at the polarization value of $P = 16$ μC cm$^{-2}$ for all samples. Figure 4b shows the inverse total capacitance ($C_{DE-AFE}$) at $P = 16$ μC cm$^{-2}$ as a function of the inverse dielectric capacitance ($C_{DE}$) with and excellent linear fit ($R^2 = 0.9986$). Two important conclusions can be drawn from Fig. 4. First, the negative intercept for $1/C_{DE} = 0$ and the consistent capacitance enhancement shows that the ZrO$_2$

capacitance must indeed be negative for all samples. Second, the ZrO$_2$ and Al$_2$O$_3$/HfO$_2$ layers do act as expected for two capacitors in series, i.e., the antiferroelectric negative capacitance is independent of the thickness of the HfO$_2$ layer. Previous results for multi-domain ferroelectric/dielectric superlattices showed a similar behavior[35]. However, theory suggests that multi-domain ferroelectric negative capacitance can strongly depend on the domain configuration and lateral domain wall motion in the ferroelectric and thus changes, e.g. with the ferroelectric film thickness[36–39]. On the other hand, the constant $C_{AFE} < 0$ of our antiferroelectric ZrO$_2$ films with both HfO$_2$ and ZrO$_2$ thickness (see also Supplementary Fig. 8) indicates that it is an intrinsic property of the non-polar to polar structural transition. This suggests that antiferroelectricity in ZrO$_2$ is indeed caused by a local field-induced inversion symmetry breaking of the unit cell. Our findings do not support the recently proposed

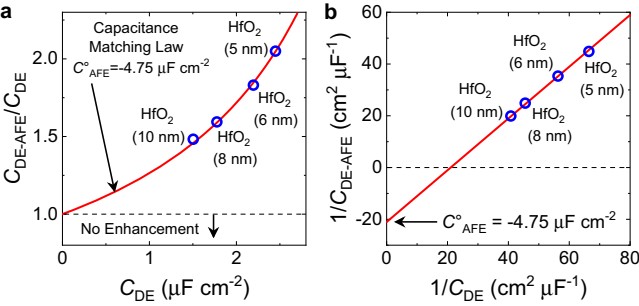

**Fig. 4 Capacitance matching in antiferroelectric-dielectric heterostructure capacitors. a** The capacitance enhancement factor $r = C_{DE\text{-}AFE}/C_{DE}$ in dielectric-antiferroelectric heterostructures with varying HfO$_2$ thickness as functions of the constituent dielectric capacitance $C_{DE}$. $C_{DE\text{-}AFE}$ is the heterostructure capacitance. The best fit to the capacitance matching law: $C_{DE\text{-}AFE}/C_{DE} = |C°_{AFE}|/(|C°_{AFE}| - C_{DE})$ is obtained for $C°_{AFE} = -4.75\,\mu F\,cm^{-2}$ at $P = 16\,\mu C\,cm^{-2}$ with $R^2 = 0.9986$, which is plotted as the red line in **a** and **b**. **b** $1/C_{DE\text{-}AFE}$ is shown as a function of $1/C_{DE}$. The intercept gives the inverse antiferroelectric capacitance $1/C°_{AFE}$, which is negative. Note that the negative capacitance of ZrO$_2$ reported here is independent of $C_{DE}$.

antiferroelectric model of domain depinning in a depolarized ferroelectric assuming a tetragonal/orthorhombic phase mixture as the ground state[40,41].

**Negative capacitance transistor simulations.** Finally, we simulated an antiferroelectric negative capacitance field-effect transistor (NCFET) based on our experimental results to investigate what performance improvements might be expected compared to current device technologies. The details can be found in the Supplementary Methods. We find that the antiferroelectric NCFET outperforms the reference device in terms of speed while at the same time consuming 41% less power at almost half the power supply voltage (see Supplementary Fig. 13 and Supplementary Table 1). These results support previous suggestions that antiferroelectrics are indeed promising for ultra-low power electronic devices[42]. Especially since ZrO$_2$ is already used as the capacitor dielectric in current dynamic random access memory technologies, which has been demonstrated to also be antiferroelectric[43].

In summary, we have explored the forbidden thermodynamic region of the non-polar to polar antiferroelectric transition in ZrO$_2$ using pulsed electrical measurements. In contrast to previous reports in multi-domain ferroelectrics, we report a different type of antiferroelectric negative capacitance which seems to be an intrinsic property of the structural instability at the boundary between P4$_2$/nmc and Pca2$_1$ phases in fluorite-type oxides. This is of great interest for applications in energy-efficient electronics since mixtures of these phases have been reported in the widely used HfO$_2$ and ZrO$_2$ based ultrathin films, which are present in most semiconductor products today. The observation of negative capacitance in ZrO$_2$ indicates that the switching pathway between the P4$_2$/nmc and Pca2$_1$ might include an intermediate saddle phase of different symmetry. Furthermore, our findings suggest that the phenomenon of negative capacitance is not limited to polar ferroelectric order but can be expected at any transition with a polarization instability. This includes phase transitions between non-polar and polar phases as shown here, but which should also include transitions between different polar phases or even between different non-polar phases[44]. This opens up opportunities for characterizing structural phase transitions and for building negative capacitance devices from a much broader range of possible materials.

## Methods

**Sample fabrication.** The atomic layer deposition (ALD) of ZrO$_2$ and bottom TiN layers were conducted in a 300 mm Tokyo Electron thin film formation tool. The ZrO$_2$ and TiN films were deposited by ALD at 350 °C and ~ 430 °C respectively by previously described processes[45]. The as-grown ZrO$_2$ layer was amorphous. With regards to the thickness of ZrO$_2$ in the ZrO$_2$/TiN heterostructure of these samples, there were two differently processed stacks: ZrO$_2$/TiN with 10 nm ZrO$_2$ and 5 nm ZrO$_2$, respectively. A post-ZrO$_2$ deposition annealing was performed on both stacks at 450 °C for 30 s in nitrogen atmosphere to stabilize the ZrO$_2$ layer in the antiferroelectric tetragonal phase. The ZrO$_2$/TiN heterostructure was cleaned in standard clean-2 (SC-2) solution. Afterwards, HfO$_2$ (6, 8, 10 nm)/Al$_2$O$_3$ (~ 1 nm) heterostructure was deposited on the ZrO$_2$/TiN heterostructure in a Cambridge NanoTech Fiji G2 Plasma Enhanced ALD (PEALD) system at Georgia Tech. Subsequently, a top TiN layer was deposited without breaking the vacuum in the same ALD reactor. The deposition was carried out at 250 °C using tetrakis (dimethylamido) hafnium, tetrakis (dimethylamido) aluminum and tetrakis (dimethylamido) titanium precursors for HfO$_2$, Al$_2$O$_3$ and TiN, respectively. For oxides, water was used as the oxygen source and for TiN, nitrogen (plasma) was used as the nitride source. After the ALD of TiN/HfO$_2$/Al$_2$O$_3$/ZrO$_2$/TiN and TiN/HfO$_2$/ZrO$_2$/TiN heterostructures, Al (~ 100 nm) was evaporated and patterned into rectangular electrodes using standard microfabrication techniques. Top TiN layer was wet etched afterwards in H$_2$O$_2$:H$_2$O solution at 50 °C using the patterned Al layer as a hard mask.

**Electrical characterization.** Electrical measurements were conducted on a Cascade Microtech Summit 1200 K Semi-automated Probe Station. Polarization versus electric field curves were measured using an aixACCT TF-3000 ferroelectric parameter analyzer at 1 kHz with dynamic leakage current compensation (DLCC) turned off. The capacitance versus electric field curves were measured by a Keysight E4990A impedance analyzer with a small-signal amplitude of 25 mV and a frequency of 100 kHz. The pulsed measurements on dielectric/antiferroelectric heterostructure capacitors were conducted using a Keysight 81150 A Pulse Function Arbitrary Noise Generator and a Keysight DSOS104A Oscilloscope. The series resistor had a resistance of $R = 5.6\,k\Omega$.

**Structural characterization.** X-ray diffraction (XRD) measurements were performed using SmartLab diffractometer (Rigaku) equipped with 9 kW cooper rotating anode X-ray source (wavelength CuKα = 0.15418 nm). A standard co-planar diffraction geometry with constant angle of incidence (grazing incidence X-ray diffraction: GIXRD) was used for the studies. To perform the GIXRD measurements the diffractometer was equipped with a parabolic multilayered X-ray mirror and a set of axial divergence eliminating soller slits with divergence of 5° in incident and diffracted beam, and parallel plate collimator with acceptance of 0.5° in diffracted beam. Diffracted intensity was acquired with hybrid-pixel single photon counting 2D detector HyPix 3000. The incidence angle of the X-rays, with respect to the sample surface, was set up slightly above the critical angle of investigated material, $\omega = 0.6°$. Measured diffraction data were fitted using whole powder pattern procedure - Rietveld method. A computer program MStruct was used for the fitting. The investigated samples contain tetragonal ZrO$_2$ phase (space group P4$_2$/nmc, #137). Some minor fractions of other ZrO$_2$ phases might be present, but in fractions below the detection limit of GIXRD. Scanning transmission electron microscopy (STEM) imaging was performed using a Hitachi aberration corrected STEM HD2700 operating at 200 kV at Georgia Tech. The convergent angle is 35 mrad, and the inner angle of the annular dark field (ADF) detector is 50 mrad. Nano beam electron diffraction (NBED) was performed using a FEI transmission electron microscope (TEM) Technai F30 working at 300 kV. The determination of the phase of ZrO$_2$ was accomplished by analyzing the NBED patterns.

**Extraction of P–E$_a$ characteristics from pulsed measurements.** Voltage pulses $V_{in}$ of duration $T = 1.1\,\mu s$ with different amplitudes were applied, and the voltage across the dielectric/antiferroelectric capacitor, $V_{DE\text{-}AFE}$ and $V_{in}$ were measured using an oscilloscope. The input pulse amplitude $V_{in}$ was changed in 200 mV steps to span the full $P–E_a$ curves. At a given applied voltage amplitude $V_a$, voltages across the dielectric layers and the antiferroelectric layer was calculated as $V_{DE} = \Delta Q/C_{DE}$ and $V_{AFE} = V_a - V_{DE}$, respectively. For each of the dielectric/antiferroelectric samples, an equivalent dielectric only stack with the same thicknesses of HfO$_2$ and Al$_2$O$_3$ (~ 1 nm) layers was deposited under the same process conditions to measure $C_{DE}$ independently (measurements shown in Supplementary Fig. 4). Figure 3d as well as Supplementary Fig. 5, 6c, 7c and Fig. 8a show the polarization $P = \Delta Q/A$ as a function of the extracted electric field in the antiferroelectric $E_a = V_{AFE}/t_a$ where $A$ and $t_a$ are the capacitor area and the ZrO$_2$ thickness. The details of the calculations are provided below.

The voltage $V_{in}$ applied to the system can be written as

$$V_{in} = IR + V_{DE-AFE} \tag{1}$$

with

$$V_{DE-AFE} = V_{DE} + V_{AFE}, \tag{2}$$

where $V_{DE\text{-}AFE}$, $V_{DE}$, and $V_{AFE}$ are the voltages across the entire dielectric/antiferroelectric heterostructure, dielectric combo, and antiferroelectric layer, respectively. The current $I(t)$ flowing through the resistor $R$ in series with the dielectric/antiferroelectric heterostructure is calculated from measured $V_{in}(t)$ and $V_{DE\text{-}AFE}(t)$ waveforms. The amount of charge $Q$ on the heterostructure capacitor is calculated by

$$Q(t) = \int I(t)dt - C_{para}V_{DE-AFE}(t), \quad (3)$$

where $C_{para}$ is the parasitic capacitance that appears in parallel to the dielectric/antiferroelectric capacitor, which was experimentally determined as ~ 20 pF. Three important charges are extracted for each value of $V_a$: the maximum stored charge on the capacitor, $Q_{max} = Q(t=T)$, the residual charge on the capacitor after the applied voltage is zero again, $Q_{res} = Q(t=5\,\mu s \gg T)$, and the charge that is reversibly stored and released from the capacitor, $\Delta Q = Q_{max} - Q_{res}$. The electric field $E_a$ in the antiferroelectric layer can be calculated by

$$E_a = \frac{1}{t_a}\left(V_a - \frac{\Delta Q}{C_{DE}}\right), \quad (4)$$

where $\Delta Q$ is the amount of charges reversed and $C_{DE}$ is the capacitance of the dielectric layer. The relative permittivity $\varepsilon_r$ of the dielectric layers was extracted from capacitance versus voltage measurements on samples fabricated without the antiferroelectric layer (see Supplementary Fig. 4). The polarization of the antiferroelectric layer was calculated as

$$P = \frac{\Delta Q}{A} - \varepsilon_0 E_a \approx \frac{\Delta Q}{A}, \quad (5)$$

where $A$ is the area of the dielectric/antiferroelectric capacitor. Using Eqs. (4) and (5), the antiferroelectric $P$ versus $E_a$ curve can be calculated. The experimental free energy density $G$ can be obtained from

$$G = \int E_a(P)dP. \quad (6)$$

**Reporting summary**. Further information on research design is available in the Nature Research Reporting Summary linked to this article.

## Data availability
The data that support the findings of this study are available from the corresponding authors upon reasonable request.

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

## Acknowledgements

This work was supported in part by the National Science Foundation through the CAREER Award (# 2047880), in part by the Global Research Collaboration (GRC) program of the Semiconductor Research Corporation (SRC) and in part by the Applications and Systems-Driven Center for Energy-Efficient Integrated Nano Technologies (ASCENT), one of six centers in the Joint University Microelectronics Program (JUMP), an SRC program sponsored by the Defense Advanced Research Program Agency (DARPA). This work was performed in part at the Georgia Tech Institute for Electronics and Nanotechnology, a member of the National Nanotechnology Coordinated Infrastructure (NNCI), which is supported by the National Science Foundation (ECCS-1542174). M.D. acknowledges the financial support from the project NanoCent-Nanomaterials Centre for Advanced Applications, Project No. CZ.02.1.01/0.0/0.0/15\_003/0000485 financed by ERDF. S.E.R.-L. acknowledges support from ANID FONDECYT Iniciación en Investigación Grant No. 11180590 and the supercomputing infrastructure of the NLHPC (ECM-02). J.R. acknowledges the support from Air Force Office of Scientific Research under award no. FA9550-16-1-0335. This work was in part financially supported out of the State budget approved by the delegates of the Saxon State Parliament.

## Author contributions

Z.W., N.T., M.H. and A.I.K. conceived the experiment. Z.W. and N.T. performed the electrical measurements. N.T. led the growth using the ALD technique and the capacitor fabrication steps at Georgia Tech. S.C. and K.T. performed ALD of $ZrO_2$/TiN heterostructure using a 300 mm Tokyo Electron thin film formation tool. M.T. performed the TEM experiments. M.D. performed the GIXRD experiments. S.S.K.P. performed the modeling of AFE NCFETs. Z.W., N.T., M.H., M.T., J.K., S.E.R.-L, J.R. and A.I.K. discussed the results and wrote the initial manuscript. A.Z., P.V.R., A.G., D.T., R.C., J.H., S.K.L., S.Y., W.C., D.A., J.R., S.S. and T.M. discussed the results and gave inputs to the final manuscript.

## Competing interests

The authors declare no competing interests.
