## [Peer review file · Nature Communications]

REVIEWER COMMENTS

Reviewer #1 (Remarks to the Author):

The present manuscript describes the first experimental demonstration of the negative capacitance effect utilizing the unconventional antiferroelectricity of zirconia. The material is antiferroelectric in the sense that it shows a double hysteresis loop in the P-E measurements. However, unlike conventional antiferroelectrics which have anti-polar alignments of atomistic dipoles, it has been claimed that the material is microscopically nonpolar and transforms to a polar phase upon bias application. This is confirmed, albeit indirectly, through the measurement of the capacitance enhancement (i.e., negative capacitance) in the antiferroelectric-dielectric stacked capacitor. The notion that phase transitions induced by applied bias can be used to induce negative capacitance is indeed exciting, and I believe that this will be of interest to the readership of Nature Communications. There are a few issues I would like to have clarified, however.

1. The authors have confirmed through structure analysis that zirconia in the as-prepared stack is tetragonal. However, I couldn't pick up any direct evidence that the antiferroelectric P-E cycling behavior results from repeated tetragonal-orthorhombic transitions. As the authors say, transitions between different polar or even nonpolar phases may give rise to this sort of behavior. It seems to be simply assumed that the structure at high voltages is the orthorhombic polar phase based on previous DFT simulations, but it would be nice if the manuscript could provide experimental evidence.
2. On a related note, it has been proposed previously using DFT simulation that an ultrathin ferroelectric film with striped domains may exhibit antiferroelectric-like behavior with negative capacitance (S. Kasamatsu et al., *Adv. Mater.* 2016, 28, 335–340). The proposed mechanism in the present manuscript is different from this, but it may be worth referring to, as it is another possible transition mechanism with antiferroelectric-like polarization instability.
3. I understand that Fig. 1 is a general description of antiferroelectricity, but the given energy landscape is far from what is reported in zirconia, so it might be misleading to the reader. Figure 1b makes it look like the polar phase is not even locally stable at $E_a=0$, but that doesn't seem to be the case if you look at DFT predictions (Ref. 24, 29). Also, there is no guarantee that M is lower than N in Fig. 1c.
4. Could the authors clarify the role of the dielectric layer? In the FE-DE literature, the role seems to be two-fold: (1) to provide a depolarizing effect so that the FE is stabilized in a near-nonpolar negative capacitance state, and (2) to provide blocking against charge injection that would facilitate polarization reversal, which destabilizes the negative capacitance state. I'm confused because in the present proposed mechanism, the negative capacitance does not come from the depolarized state, and also there is no polarization reversal.

Reviewer #2 (Remarks to the Author):

The authors report experimental evidence on the negative capacitance during the phase transition in ZrO₂ and show that it arises from the intrinsic property of the structural instability. The topic is novel and of technological importance for the electronic devices. However, there are some questions that should be clarified and major revision is required before the publication of this manuscript.

1. Regarding Fig.3d, there are some questions need to be clarified.
 - a. The theoretical prediction of the negative capacitance in Fig.1a shows a curve consisting of two S-shapes on both the positive and negative sides. However, the experimental data shown in Fig.3d do not form a complete S-shape on both sides. Why are the data of higher electric field missing? The authors should give an explanation.
 - b. In Fig.3d, the P-E curve of the pure AFE ZrO₂ layer is measured by a ferroelectric test analyzer, which usually applies an excitation voltage of triangular waveform with the frequency below 10 kHz.

However, the P-E curve of the DE/AFE stack is measured by microsecond voltage pulses. Thus the direct comparison of the two P-E curves may be inappropriate.

2. Regarding Fig.4, there are critical points that need to be addressed.

- a. What does the "average capacitance enhancement" in the line 213-214 of the main test mean? Does it mean that the capacitance enhancement is averaged over the voltage region of the negative capacitance? If so then it is questionable that the average capacitance enhancement can be used to the ideal capacitance matching equation described in line 215, since the negative capacitance in AFE is not a constant and depends on the applied voltage according to the theoretical prediction in Fig.1a.
- b. The author should add some extra data points in Fig.4 to make the fitting analysis more convincing.

3. The authors used the series resistor R to measure the current I(t) flowing through the dielectric/antiferroelectric heterostructure as calculated from the difference between the measured $V_{in}(t)$ and $V_{DE-AFE}(t)$ waveforms. This method is not rigorous and very questionable because the series resistor R causes a significant influence on the response time and the characteristics of the tested system. In order to get more convincing data, additional experiments need be performed by using at least two different series resistor R (of different orders of magnitude) to check whether similar phenomena exist or not. Actually, the current I(t) flowing through the tested device can be directly measured with a high precision by advanced instruments such as Keithley 4200 SCS or Keysight B1500A without using the series resistor R. It is strongly recommended not to use the series resistance R for the current measurement. The author should provide the experiment data without using the series resistance R to reduce the influence of R on the response time and the characteristics of the measured system.

4. The ref.29 mentioned in Fig.4 should be ref.31.

Reviewer #3 (Remarks to the Author):

Unfortunately, I cannot recommend the manuscript for publication at this point. First, the reported research does not look original enough to justify publication in the Nature group magazines. Second, the conclusions supposedly drawn from the experimental observations are not sufficiently supported by the reported experimental data. In more detail, the reasons are as follows.

The major point is that the authors claim the observation of the negative capacitance induced by the field-triggered switch between the two structural phases, presumably anti-ferroelectric and ferroelectric ones. However, this phenomenon has already been observed in 1969 by Vogel R, Walsh PJ [Appl. Phys. Lett, 14, 216] and is well known to the semiconducting community. Thus, although the authors' statement that the "negative capacitance is a more general phenomenon than previously thought and can be expected in a much broader range of materials exhibiting structural phase transitions" seems to be 200% correct, it is not clear how does it prove that what is observed in this manuscript is indeed a stable "negative capacitance."

The point of confusion is that the phenomenon discussed by the authors and called them the "negative capacitance" is, strictly speaking, not the one. According to the textbook definition, the capacitance is given by the relation $C=V/Q$ where Q is the total charge at the electrodes. But what the authors observed is the local negative slope in the V(Q) dependence, $dV/dQ<0$, occurring well far from the $Q=0$ point. This quantity is conventionally called the differential negative capacitance and appears in a rich multitude of manifestations, for example, in the mentioned above publication by Vogel R and Walsh PJ.

On the technical side, there are several prime importance flaws and insufficiently grounded statements.

1) The authors use the Kittel model of the ferroelectric-antiferroelectric transition to describe the experimentally observed hysteresis. However, even a brief comparison of the theoretical hysteresis plot (Fig.1) with the experimental observations (Fig.2) shows that these are entirely different phenomena. Neither a single trace of the first-order phase transition nor an expected high-field saturation of the curve is observed experimentally. The experimentally observed "forbidden" branch, shown in Fig. 3d, starts at the point where the P(E) linearity loss occurs, which corresponds to the point M of the theoretical plot, rather than to the instability point B. Moreover, the "forbidden branch" does not join the high-field (paraelectric) branch, but goes somewhere else (Fig. S6c). To justify the theoretical approach, the authors should present a theoretical fit of the experimental curve, based, in particular, on the given in SI S2 fitting coefficients, to justify the used parameters, and convincingly explain the apparent discrepancy between the data and the theory.

2) The experimental foundation of the work, in particular the characterization of the hysteretic phenomena, needs further clarification. The absence of the saturation point and the profound hysteresis raises the question about its thermodynamic origin, in particular, whether the low-field and high-field branches are locally reversible? Authors should use the additional protocols of the field sweep with different sweep-inversion points to establish that both branches are stable and unique and that no other history-dependent hysteresis paths arise and interfere in the P-E space. Only in this case, the attribution of the hysteresis branches to particular phases and the thermodynamic consideration of the NC would make sense.

3) The possible role of domains of different types that easily arise in films, especially in the AFE phase is not clarified in the article. Even the possibility of the presence of domains of different types is not verified experimentally. To formulate the description of domains resting on the Kittel theory, the authors should experimentally demonstrate that their hysteresis curve is due to a single-phase transformation rather than being caused by the collective domain switching. The microscopy-scale pictures of the domain pattern and the field dependencies of domain configurations should be presented.

4) All the above points are critical for achieving an unambiguous interpretation of the assumed observation of the differential NC in the "forbidden region". To summarize: (i) The experimental data, presented in Fig. 3 are insufficient to judge whether the supposed NC "forbidden" branch is reversible. To clarify the reversibility, the authors should use the reverse protocols of the gradual charge removing from the highest charge-carrying state and demonstrate that the "forbidden" hysteretic branch returns back along the same thermodynamic path. Furthermore, to confirm the S-curve description, the authors should reveal experimentally, how their "forbidden branch" joins the high-field stable hysteretic branch. (ii) To build on their statement that the (differential) NC effect is caused by the structural phase transition, the authors should either experimentally demonstrate that the system is in a single-domain state or, if this is not the case, to clarify the impact of a many-domain structure on the assumed phenomena manifestations.

Minor comments:

5) Since the differential NC effect takes place at some finite (working) voltage it will be better to indicate the real values of the voltage in Figs. S9, rather than the relative ones. It is also advisable to indicate in Supplementary S2, the value of such working voltage for the modeled transistor and whether such values are operational in modern nanoelectronics. A broad readership would benefit from having access to direct data.

6) Many key statements of the Manuscript are formulated in a vague and fogging manner. For example:

- "An applied electric field of around 2-3 MV/cm can then transform the non-polar tetragonal phase into the polar orthorhombic Pca21 phase, which is known to be ferroelectric [26]. While obtaining definitive experimental proof of such a field-induced first-order phase transition has proved difficult so far [27], this mechanism is consistent with first-principles calculations [24] [28] as well as composition- and temperature-dependent experimental results [25]. While the microscopic switching pathway between the P42/nmc and Pca21 phases is still unclear, it has been suggested to include an intermediate phase of orthorhombic Pmn21 symmetry for Hf_{0.5}Zr_{0.5}O₂ [29]."

Authors should indicate clearly, whether the ferroelectric nature of the high-field phase is proven or this is just their hypothesis not actually confirmed by the experiment. Reference [26] is misleading. The statement that "the polar orthorhombic Pca21 phase, ... is known to be ferroelectric" is not the achievement of Ref. [26] but is the subject of the textbook knowledge. The way the reference is given provokes the feeling that Ref. [26] proves that the high-field phase observed in the present manuscript is ferroelectric as well.

- "Supplementary Fig. S1: Grazing-incidence X-ray diffraction measurement of ZrO₂. Diffraction patterns and their indices for tetragonal ZrO₂ as well as TiN and Al phases are marked in the figure. For comparison, the position of the diffraction peaks of orthorhombic ZrO₂ and monoclinic ZrO₂ are indicated at the bottom. The Bragg peaks for ZrO₂ match well with those of the tetragonal structure. No resemblance of the diffraction patterns of our samples with the orthorhombic and monoclinic patterns is observed in these samples indicating that the crystalline phase in these samples are predominantly tetragonal with negligible fractions (below the detection limit of our XRD set-up) of orthorhombic and monoclinic phases".

The difference in patterns and the absence of the orthorhombic or monoclinic phases are different statements. The authors should indicate clearly, whether the fractions of orthorhombic and monoclinic phases were observed or not.

The authors report experimental evidence on the negative capacitance during the phase transition in ZrO₂ and show that it arises from the intrinsic property of the structural instability. The topic is novel and of technological importance for the electronic devices. However, there are some questions that should be clarified and major revision is required before the publication of this manuscript.

1. Regarding Fig.3d, there are some questions need to be clarified.

a. The theoretical prediction of the negative capacitance in Fig.1a shows a curve consisting of two S-shapes on both the positive and negative sides. However, the experimental data shown in Fig.3d do not form a complete S-shape on both sides. Why are the data of higher electric field missing? The authors should give an explanation.

b. In Fig.3d, the P-E curve of the pure AFE ZrO₂ layer is measured by a ferroelectric test analyzer, which usually applies an excitation voltage of triangular waveform with the frequency below 10 kHz. However, the P-E curve of the DE/AFE stack is measured by microsecond voltage pulses. Thus the direct comparison of the two P-E curves may be inappropriate.

2. Regarding Fig.4, there are critical points that need to be addressed.

a. What does the “average capacitance enhancement” in the line 213-214 of the main test mean? Does it mean that the capacitance enhancement is averaged over the voltage region of the negative capacitance? If so then it is questionable that the average capacitance enhancement can be used to the ideal capacitance matching equation described in line 215, since the negative capacitance in AFE is not a constant and depends on the applied voltage according to the theoretical prediction in Fig.1a.

b. The author should add some extra data points in Fig.4 to make the fitting analysis more convincing.

3. The authors used the series resistor R to measure the current $I(t)$ flowing through the dielectric/antiferroelectric heterostructure as calculated from the difference between the measured $V_{in}(t)$ and $V_{DE-AFE}(t)$ waveforms. This method is not rigorous and very questionable because the series resistor R causes a significant influence on the response time and the characteristics of the tested

system. In order to get more convincing data, additional experiments need be performed by using at least two different series resistor R (of different orders of magnitude) to check whether similar phenomena exist or not. Actually, the current $I(t)$ flowing through the tested device can be directly measured with a high precision by advanced instruments such as Keithley 4200 SCS or Keysight B1500A without using the series resistor R . It is strongly recommended not to use the series resistance R for the current measurement. The author should provide the experiment data without using the series resistance R to reduce the influence of R on the response time and the characteristics of the measured system.

4. The ref.29 mentioned in Fig.4 should be ref.31.

Response to Reviewers

Reviewer #1:

R1: The present manuscript describes the first experimental demonstration of the negative capacitance effect utilizing the unconventional antiferroelectricity of zirconia. The material is antiferroelectric in the sense that it shows a double hysteresis loop in the P-E measurements. However, unlike conventional antiferroelectrics which have anti-polar alignments of atomistic dipoles, it has been claimed that the material is microscopically nonpolar and transforms to a polar phase upon bias application. This is confirmed, albeit indirectly, through the measurement of the capacitance enhancement (i.e., negative capacitance) in the antiferroelectric-dielectric stacked capacitor. The notion that phase transitions induced by applied bias can be used to induce negative capacitance is indeed exciting, and I believe that this will be of interest to the readership of Nature Communications. There are a few issues I would like to have clarified, however.

Response: We thank the reviewer for this overall positive assessment. We will address the remaining issues in the following.

R1: 1. The authors have confirmed through structure analysis that zirconia in the as-prepared stack is tetragonal. However, I couldn't pick up any direct evidence that the antiferroelectric P-E cycling behavior results from repeated tetragonal-orthorhombic transitions. As the authors say, transitions between different polar or even nonpolar phases may give rise to this sort of behavior. It seems to be simply assumed that the structure at high voltages is the orthorhombic polar phase based on previous DFT simulations, but it would be nice if the manuscript could provide experimental evidence.

Response: We agree with the reviewer. Direct experimental evidence of the tetragonal to orthorhombic phase transition would be the ultimate proof. However, experimentally, this has proved to be very difficult to achieve. Observation of the transition in TEM with *in situ* application of an electric field is hard, since a very high resolution is necessary to clearly distinguish the phases (ideally imaging the oxygen atoms), and one also needs a grain with a good zone axis. Some of the authors of this manuscript have tried this before (see DOI: 10.1109/VLSITechnology18217.2020.9265091) but could not conclusively show the tetragonal to orthorhombic transition, since the position of the oxygen atoms could not be resolved. However, very recently, another group has directly shown for the first time that the tetragonal phase can transform into the orthorhombic phase under application of an electric field in $\text{Hf}_{0.5}\text{Zr}_{0.5}\text{O}_2$ (Y. Zheng et al., "*In-situ atomic visualization of structural transformation in $\text{Hf}_{0.5}\text{Zr}_{0.5}\text{O}_2$ ferroelectric thin film: from nonpolar tetragonal phase to polar orthorhombic phase,*" 2021 Symposium on VLSI Technology, 2021, pp. 1-2.). It seems very likely that similar transformations occur in our ZrO_2 films.

In principle, it might be possible that the transition in ZrO_2 is not directly from the tetragonal to the orthorhombic $\text{Pca}2_1$ phase, but from the tetragonal phase to a different polar phase. However, we are certain that the phase at high electric field must be polar, due to the P-V loop in Figure 2a. Such a behavior cannot be explained by a non-polar to non-polar transition since the slope dP/dV of the loop at zero volts and at the highest voltages (in saturation) is nearly identical. A P-V hysteresis for non-polar to non-polar transitions is only possible if there is a

large difference in the permittivity between both phases, which we did not observe. Therefore, our structural data demonstrating the non-polar tetragonal phase in the ground state in combination with the electrical observation of the double-hysteresis loop can only be explained by a non-polar to polar field-induced structural phase transition.

Although there is no direct evidence that the polar phase is of orthorhombic $Pca2_1$ symmetry, there is substantial indirect evidence: 1) It has been shown in many publications that when increasing the temperature or changing composition (e.g. decreasing x from 1 in the $Hf_{1-x}Zr_xO_2$ solid solution), an orthorhombic to tetragonal phase transition is observed (see e.g. <https://doi.org/10.1063/5.0002835>). Furthermore, there have been reports of ferroelectric orthorhombic ZrO_2 layers under certain processing conditions. Lastly, the field-induced tetragonal to orthorhombic transition has been directly observed in $Hf_{0.5}Zr_{0.5}O_2$ (see ref. in the first paragraph), which has many structural similarities with ZrO_2 .

These experimental findings suggest that the orthorhombic phase can indeed be stabilized in ZrO_2 thin films, although they tend to crystallize in the tetragonal phase in most cases. Furthermore, they show that the transition between the tetragonal and orthorhombic phase in fluorite structure oxides can be triggered by temperature, composition, and electric field – consistent with DFT calculations. To us, the combination of all these experimental and theoretical results presents strong indirect evidence that our antiferroelectric ZrO_2 films undergo a tetragonal to orthorhombic $Pca2_1$ phase transition with applied electric field. The direct experimental confirmation of symmetry of the polar phase under high applied field in ZrO_2 is an important task for future research that we feel is beyond the scope of the present manuscript.

We have modified the text on page 4, to make this point clearer to the reader. First, we changed the following sentences explaining the current understanding of the antiferroelectric transition in ZrO_2 :

“The current understanding of the origin of antiferroelectricity in ZrO_2 is that the non-polar tetragonal phase undergoes a first-order structural phase transition into the polar orthorhombic $Pca2_1$ phase by application an electric field of around 2-3 MV/cm [25] [26]. The polar orthorhombic $Pca2_1$ phase has been shown to be responsible for the ferroelectric behavior observed in HfO_2 based thin films [27].”

Furthermore, we added additional references and explanations for why there must be a non-polar to polar transition in our films, and why we believe that the polar phase is of $Pca2_1$ symmetry:

“Since the structural data suggests that our ZrO_2 films are fully tetragonal without applied voltage, the measured double hysteresis loops can only be explained by a field-induced structural transformation into a polar ferroelectric phase. While we cannot directly determine the symmetry of this polar phase at high electric field, there is substantial evidence in literature which suggests that it is of orthorhombic $Pca2_1$ symmetry [26]. For example, a temperature-dependent phase transition from the non-polar $P4_2/nmc$ phase to the polar $Pca2_1$ phase in similar ZrO_2 thin films has been experimentally observed with in situ with high-temperature X-ray diffraction [31]. Furthermore, ZrO_2 can be stabilized in the $Pca2_1$ phase even at room temperature under certain processing conditions [32]. Lastly, the field-induced transition from the $P4_2/nmc$ phase to the $Pca2_1$ phase has been directly observed in $Hf_{0.5}Zr_{0.5}O_2$ thin films [33]. From these experimental data and previous first principles calculations [25] [29], it seems

reasonable to conclude that the polar phase observed at high electric field in Fig. 2a is of $Pca2_1$ symmetry.”

R1: 2. On a related note, it has been proposed previously using DFT simulation that an ultrathin ferroelectric film with striped domains may exhibit antiferroelectric-like behavior with negative capacitance (S. Kasamatsu et al., Adv. Mater. 2016, 28, 335–340). The proposed mechanism in the present manuscript is different from this, but it may be worth referring to, as it is another possible transition mechanism with antiferroelectric-like polarization instability.

Response: We thank the reviewer for pointing this out and agree that in the context of our manuscript, this shows another mechanism for negative capacitance from an antiferroelectric-like polarization instability. We have added the reference on page 3:

“In a similar way, first principles calculations of ferroelectric/paraelectric heterostructures have predicted antiferroelectric-like negative capacitance regions due to the transition between a striped domain pattern and a ferroelectric monodomain state [24].”

R1: 3. I understand that Fig. 1 is a general description of antiferroelectricity, but the given energy landscape is far from what is reported in zirconia, so it might be misleading to the reader. Figure 1b makes it look like the polar phase is not even locally stable at $E_a=0$, but that doesn't seem to be the case if you look at DFT predictions (Ref. 24, 29). Also, there is no guarantee that M is lower than N in Fig. 1c.

Response: We appreciate this comment. Indeed, our goal with this figure was to provide a qualitative and intuitive understanding of where antiferroelectric negative capacitance comes from. However, since Fig. 1 is based on a thermodynamic free energy model, it cannot be directly compared to the energy landscapes obtained from first principles calculations, which are computed at 0 K. Importantly, when we want to obtain an antiferroelectric double hysteresis loop as in Fig. 1a without remanent polarization from a thermodynamic model, it is necessary that the Gibbs free energy G in Fig. 1b only has a single minimum at polarization $P = 0$. The reason for this is as follows:

For simplicity, let us consider an arbitrary energy landscape as

$$G = aP^2 + bP^4 + cP^6 - EP,$$

where a, b and $c > 0$ are the thermodynamic coefficients. The equilibrium polarization P for a given electric field E , can then be found by minimizing G with respect to P , i.e., setting $dG/dP = 0 = 2aP + 4bP^3 + 6cP^5 - E$. Now if G would have multiple energy minima at $E = 0$, then there would be multiple solutions to the equation $0 = 2aP + 4bP^3 + 6cP^5$ besides the trivial solution at $P = 0$, which means that the material has two possible spontaneous polarization states at $E = 0$. To illustrate this point, in Figure R1 we have plotted an example for a triple well energy-landscape (at $E = 0$) and the corresponding P - E curve for the parameters $a = 1e9$, $b = -1e11$, $c = 3e12$:

Figure R1. Example for a triple-well free energy landscape (right) and corresponding polarization-electric field relationship (left).

Note that points 'A' and 'B' correspond to the additional energy minima of G . This means that for an antiferroelectric, where at $E = 0$ we can only have the trivial solution $P = 0$, the energy landscape for $E = 0$ can only have a single energy minimum.

Regarding the case for a non-zero applied field in Figure 1c, we agree that the minimum at point N can be lower than that of M for higher applied fields or different thermodynamic coefficients. To make this point clearer, we added the following sentence on page 2:

“Note that the relative energies of the non-polar and polar states in Fig. 1c,d are dependent upon the antiferroelectric material and also the magnitude of the electric field, i.e. the polar phase will become lower in energy than the non-polar phase for even higher applied fields.”

R1: 4. Could the authors clarify the role of the dielectric layer? In the FE-DE literature, the role seems to be two-fold: (1) to provide a depolarizing effect so that the FE is stabilized in a near-nonpolar negative capacitance state, and (2) to provide blocking against charge injection that would facilitate polarization reversal, which destabilizes the negative capacitance state. I'm confused because in the present proposed mechanism, the negative capacitance does not come from the depolarized state, and also there is no polarization reversal.

Response: This is an interesting question, which is closely related to one of the main findings of our work: that negative capacitance can appear from any polarization instability. The role of the dielectric layer in our experiments is very similar to the FE/DE case: preventing charge injection and creating a depolarizing field ($E_{\text{dep}} \propto -P$) in the ZrO_2 , such that the overall electric field $E = E_{\text{dep}} + E_{\text{ext}}$ in ZrO_2 can decrease while we increase the external voltage, when E_{dep} dominates the response of E . This means that when $\Delta E_{\text{ext}} > 0$, resulting in $\Delta P > 0$, the change in depolarization field $\Delta E_{\text{dep}} < 0$ will lead to $\Delta E = \Delta E_{\text{dep}} + \Delta E_{\text{ext}} < 0$. The difference is, as the reviewer has correctly noticed, that the antiferroelectric is not in a completely depolarized state in the negative capacitance region but has a significant macroscopic polarization. This shows that the origin of NC is not necessarily the depolarized state at $P = 0$, but any polarization region that would be unstable in isolation such that $\Delta P / \Delta E < 0$. In an antiferroelectric, this instability appears at higher electric fields and non-zero polarization.

While it is true that there is no “polarization reversal” in the traditional sense in our experiments, the average polarization does still change significantly. This is analogous to a paraelectric

(tetragonal phase) which is cooled down below the Curie-temperature T_C to transform into a ferroelectric (orthorhombic phase). We start in the $P = 0$ state and below T_C the paraelectric state becomes unstable which leads to a spontaneous polarization of the material. The instability of the polarization around $P = 0$ below T_C is the origin of the negative capacitance. The difference in our case is that we promote the phase transition by applying an electric field. Since we need a finite electric field to induce the phase transition, we automatically polarize the ZrO_2 before the transition happens. Therefore, the polarization instability cannot appear at $P = 0$, but must appear at finite polarization.

We added a sentence on page 4 to make the role of the dielectric layer clearer:

“The dielectric layer has two important functions: It prevents the injection of compensating charge which could screen the bound polarization P and it creates a depolarization field in the antiferroelectric which is antiparallel to P , such that E_a can decrease while P is increasing.”

Reviewer #2:

R2: The authors report experimental evidence on the negative capacitance during the phase transition in ZrO_2 and show that it arises from the intrinsic property of the structural instability. The topic is novel and of technological importance for the electronic devices. However, there are some questions that should be clarified and major revision is required before the publication of this manuscript.

Response: Thank you for this overall positive feedback. In the following, we will address the remaining questions and explain the revisions made to the manuscript.

R2: 1. Regarding Fig.3d, there are some questions need to be clarified.

a. The theoretical prediction of the negative capacitance in Fig.1a shows a curve consisting of two S-shapes on both the positive and negative sides. However, the experimental data shown in Fig.3d do not form a complete S-shape on both sides. Why are the data of higher electric field missing? The authors should give an explanation.

Response: This is an excellent question. The short answer is that the large polarization in the antiferroelectric layer ($P > 15 \mu\text{C}/\text{cm}^2$) at high voltage, creates an additional electric field $E_{d,pol}$ across the dielectric layers in the same direction as the externally applied field $E_{d,ext}$, resulting in hard dielectric breakdown. Considering an effective dielectric of thickness t_d and permittivity ϵ_d (equivalent to the $\text{Al}_2\text{O}_3/\text{HfO}_2$ series stack), the total electric field in this dielectric is given by

$$E_d = E_{d,pol} + E_{d,ext} = \frac{P + \frac{\epsilon_0 \epsilon_{AFE}}{t_{AFE}} V_{ext}}{\epsilon_0 \left(\epsilon_d + \epsilon_{AFE} \frac{t_d}{t_{AFE}} \right)}$$

When we put our experimental parameters into this equation, we can estimate that $E_d > 10$ MV/cm for the highest applied voltages, which leads to hard dielectric breakdown of the capacitor. Similar behavior has been observed for ferroelectric/dielectric heterostructures, where often the second positive capacitance branch is not observed due to hard dielectric breakdown (see e.g. <https://doi.org/10.1038/s41586-018-0854-z>). However, on some of our fabricated

capacitors, which had a slightly better than average breakdown field strength, we were able to observe the second positive capacitance branch just before hard breakdown occurred (see Fig. R2). We added these data to the Supplementary information to confirm that we can indeed observe the second positive branch at high voltage if the sample has a better than average breakdown field strength.

Figure R2. Measurement of the second positive capacitance branch for high applied voltages. The measurement of the high polarization regions is limited by hard dielectric breakdown.

We added the following discussion to the manuscript on page 7:

“However, the second positive capacitance branch expected at higher fields is often not observed in experiments, since hard dielectric breakdown of the dielectric layers occurs at these high electric fields. Nevertheless, we were able to observe the second positive capacitance branch in some samples that showed a slightly higher than average breakdown field strength as can be seen in the Supplementary Information Fig. S9.”

R2: b. In Fig.3d, the P-E curve of the pure AFE ZrO₂ layer is measured by a ferroelectric test analyzer, which usually applies an excitation voltage of triangular waveform with the frequency below 10 kHz. However, the P-E curve of the DE/AFE stack is measured by microsecond voltage pulses. Thus the direct comparison of the two P-E curves may be inappropriate.

Response: We do agree that there are major differences between the regular triangular measurement and the pulsed measurement on the DE/AFE stacks. Indeed, when we first compared the P-E curves obtained from both measurements we were surprised by how well they matched. But when we thought about it more, it did make sense to us: If the intrinsic polarization response of the ZrO₂ layer is much faster than the one microsecond used in the pulsed measurements, it is expected that the non-polar to polar transition should happen at similar electric fields and polarization values. Since it has been shown that HfO₂ and ZrO₂ based (anti)ferroelectrics have characteristic switching times on the order of ten nanoseconds (e.g., see <https://doi.org/10.1063/1.5098786>), it therefore seems reasonable to expect a similar antiferroelectric response between measurements of 1 kHz and 1 MHz effective frequency.

The actual main difference between these two methods is that in the pulsed measurement on the DE/AFE stack, free charges cannot effectively screen the polarization in the ZrO₂, which results in our observation of negative capacitance instead of a regular P-V loop. This is consistent with previous experimental findings (<https://doi.org/10.1038/s41586-018-0854-z>).

R2: 2. Regarding Fig.4, there are critical points that need to be addressed.

a. What does the “average capacitance enhancement” in the line 213-214 of the main text mean? Does it mean that the capacitance enhancement is averaged over the voltage region of the negative capacitance? If so then it is questionable that the average capacitance enhancement can be used to the ideal capacitance matching equation described in line 215, since the negative capacitance in AFE is not a constant and depends on the applied voltage according to the theoretical prediction in Fig.1a.

b. The author should add some extra data points in Fig.4 to make the fitting analysis more convincing.

Response: We thank the reviewer for bringing up this important point. Indeed, we agree with the reviewer that our previous method of averaging the capacitance enhancement over the voltage region is not ideal due to the non-linearity. Therefore, we have changed the extraction and now only take the capacitance enhancement at $P = 16 \mu\text{C}/\text{cm}^2$ (in the middle of the NC region) for all samples. Since the P-E characteristics of the ZrO_2 should not depend on the thickness of the other dielectrics, this extraction should be robust and fit to the capacitance matching equation. Furthermore, we have fabricated and measured another sample with an even thinner layer of 5 nm HfO_2 , which we also added to Fig. 4. And indeed, using the new capacitance enhancement extraction, all four HfO_2 thicknesses can be fitted to the capacitance matching equation as shown in the revised Fig. 4.

Lastly, during the revision the manuscript, we noticed a small typo in the calculation of the ferroelectric negative capacitance reported for $\text{Pb}_{0.5}\text{Sr}_{0.5}\text{TiO}_3/\text{SrTiO}_3$ superlattices taken from reference <https://doi.org/10.1038/nature17659>, which were shown in the original Fig. 4c. After the correction, the $\text{Pb}_{0.5}\text{Sr}_{0.5}\text{TiO}_3/\text{SrTiO}_3$ data does not show a consistent trend with the dielectric capacitance, which is why we decided to remove it in the revised Fig. 4. As a result, we further changed Fig. 4, such that the data is less redundant. Fig. 4a still shows the capacitance enhancement as a function of the dielectric capacitance, while Fig. 4b now shows the total inverse capacitance of our heterostructure as a function of the inverse dielectric capacitance, which directly shows that the antiferroelectric capacitance is negative from the intercept of the linear fit. The clear linear trend in Fig. 4b also directly implies that the antiferroelectric negative capacitance is constant with C_{DE} , which is why the original Fig. 4c has been omitted.

Figure R3. (Revised Fig. 4.). The capacitance enhancement was now taken at one defined polarization value ($P = 16 \mu\text{C}/\text{cm}^2$) for all samples. The new fit of the antiferroelectric capacitance is shown as the red line. A new sample with 5 nm HfO₂ was fabricated, characterized, and added to the figure. Fig. 4c was removed for more clarity and less redundancy.

We revised the text on page 7-8 accordingly:

“Next, we changed the HfO₂ thickness to 5 nm, 6 nm and 10 nm while keeping Al₂O₃ and ZrO₂ layer thicknesses constant (~ 1 nm and 10 nm, respectively). For all different HfO₂ thicknesses, the extracted P - E_a characteristics of the ZrO₂ layer have the same quantitative shape as shown in Supplementary Fig. S5-S7. We further observed in Fig. 4a that the capacitance enhancement ($r = C_{DE-AFE}/C_{DE} > 1$) in these samples obeys the ideal capacitance matching equation $r = |C_{AFE}| / (|C_{AFE}| - C_{DE})$ with a best fit antiferroelectric negative capacitance $C_{AFE}^\circ = -4.75 \mu\text{F}/\text{cm}^2$, extracted at the polarization value of $P = 16 \mu\text{C}/\text{cm}^2$ for all samples. Fig. 4b shows the inverse total capacitance (C_{DE-AFE}) at $P = 16 \mu\text{C}/\text{cm}^2$ as a function of the inverse dielectric capacitance (C_{DE}) with an excellent linear fit ($R^2 = 0.9986$). Two important conclusions can be drawn from Fig. 4. First, the negative intercept for $1/C_{DE} = 0$ and the consistent capacitance enhancement shows that the ZrO₂ capacitance must indeed be negative for all samples. Second, the ZrO₂ and Al₂O₃/HfO₂ layers do act as expected for two capacitors in series, i.e., the antiferroelectric negative capacitance is independent of the thickness of the HfO₂ layer. Previous results for multi-domain ferroelectric/dielectric superlattices showed a similar behavior [35]. However, theory suggests that multi-domain ferroelectric negative capacitance can strongly depend on the domain configuration and lateral domain wall motion in the ferroelectric and thus changes, e.g. with the ferroelectric film thickness [36] [37] [38] [39].”

Furthermore, we changed the caption of Fig. 4 accordingly:

“**Capacitance matching in antiferroelectric-dielectric heterostructure capacitors.** a, The capacitance enhancement factor $r = C_{DE-AFE}/C_{DE}$ in dielectric-antiferroelectric heterostructures with varying HfO₂ thickness as functions of the constituent dielectric capacitance C_{DE} . C_{DE-AFE} is the heterostructure capacitance. The best fit to the capacitance matching law: $C_{DE-AFE}/C_{DE} = |C_{AFE}^\circ| / (|C_{AFE}^\circ| - C_{DE})$ is obtained for $C_{AFE}^\circ = -4.75 \mu\text{F}/\text{cm}^2$ at $P = 16 \mu\text{C}/\text{cm}^2$ with $R^2 = 0.9986$, which is plotted as the red line in a and b. b, $1/C_{DE-AFE}$ is shown as a function of $1/C_{DE}$. The intercept gives the inverse antiferroelectric capacitance $1/C_{AFE}^\circ$, which is negative. Note that the negative capacitance of ZrO₂ reported here is independent of C_{DE} .”

R2: 3. The authors used the series resistor R to measure the current I(t) flowing through the

dielectric/antiferroelectric heterostructure as calculated from the difference between the measured $V_{in}(t)$ and $V_{DE-AFE}(t)$ waveforms. This method is not rigorous and very questionable because the series resistor R causes a significant influence on the response time and the characteristics of the tested system. In order to get more convincing data, additional experiments need be performed by using at least two different series resistor R (of different orders of magnitude) to check whether similar phenomena exist or not. Actually, the current $I(t)$ flowing through the tested device can be directly measured with a high precision by advanced instruments such as Keithley 4200 SCS or Keysight B1500A without using the series resistor R . It is strongly recommended not to use the series resistance R for the current measurement. The author should provide the experiment data without using the series resistance R to reduce the influence of R on the response time and the characteristics of the measured system.

Response: We thank the reviewer for this suggestion. We are aware that the resistor normally is not needed to measure the current flowing through the device. However, for the short-pulsed measurements carried out here, equipment like the 4200 SCS or B1500A did not satisfy our need for a high time resolution to obtain reliable data. Indeed, we have tried such approaches in the past and noticed that the maximum possible number of measurement points per pulse was too small. Therefore, we switched to the oscilloscope-based setup reported in the manuscript, which has also been successfully applied by other groups in the past (e.g., see <https://doi.org/10.1021/acs.nanolett.6b01480>). However, we agree with the reviewer that we should verify that a change in the resistance R does not change the general result of observing negative capacitance in the ZrO_2 layer. Therefore, as the reviewer suggested, we have carried out additional experiments using a resistance of $R = 560 \Omega$ instead of $5.6 \text{ k}\Omega$. The results are shown in Supplementary Fig. S10 and Fig. R4. As can be seen, the negative capacitance regions are still observed although much faster pulses ($\sim 300 \text{ ns}$) were used. This is consistent with previous findings which showed that changing the timescale in such measurements does not affect the general capacitance enhancement and negative capacitance as long as they are fast enough to minimize charge trapping (e.g., see <https://doi.org/10.1109/IEDM.2018.8614677>).

Figure R4. Measurement data for an 8 nm HfO₂/1 nm Al₂O₃/10 nm ZrO₂ capacitor with a series resistance $R = 560 \Omega$.

R2: 4. The ref.29 mentioned in Fig.4 should be ref.31.

Response: Thank you for pointing out this typo. Since we decided to remove Fig. 4c as explained above, the reference did not need to be changed.

Reviewer #3:

R3: Unfortunately, I cannot recommend the manuscript for publication at this point. First, the reported research does not look original enough to justify publication in the Nature group magazines. Second, the conclusions supposedly drawn from the experimental observations are not sufficiently supported by the reported experimental data. In more detail, the reasons are as follows.

Response: We thank the reviewer for the honest and constructive feedback. It has helped us to significantly improve our manuscript. However, we feel that some of the reviewer's comments might be based on misconceptions about our work, which we want to resolve in the following.

R3: The major point is that the authors claim the observation of the negative capacitance induced by the field-triggered switch between the two structural phases, presumably anti-

ferroelectric and ferroelectric ones. However, this phenomenon has already been observed in 1969 by Vogel R, Walsh PJ [Appl. Phys. Lett, 14, 216] and is well known to the semiconducting community. Thus, although the authors' statement that the "negative capacitance is a more general phenomenon than previously thought and can be expected in a much broader range of materials exhibiting structural phase transitions" seems to be 200% correct, it is not clear how does it prove that what is observed in this manuscript is indeed a stable "negative capacitance."

Response: We thank the reviewer for bringing this paper to our attention. However, we disagree with the interpretation and conclusion presented by the reviewer. The publication in question from Vogel and Walsh, which does report a negative capacitance, is not caused by a structural phase transition. The material which was investigated is an amorphous semiconductor chalcogenide namely $\text{Te}_{0.49}\text{As}_{0.33}\text{Ge}_{0.06}\text{Si}_{0.03}\text{Ga}_{0.09}$ which belongs to the wider class of ovonic threshold switches (OTS). The current understanding of the threshold switching mechanism in such OTS devices is that it is of electronic nature and not caused by a structural phase transition:

"Since the discovery of the threshold-switching effect in 1968 by Ovshinsky several different physical mechanisms have been proposed for its explanation. An extensive discussion about whether the driving force for this reversible switching phenomenon is controlled by a thermal or by an electronic effect was settled in the 1980s in favor of an electronic excitation mechanism." (Menzel et al. Adv. Funct. Mater. 25, 6306-6325 (2015))

It is thus currently understood that OTS devices remain amorphous, even during threshold switching (see e.g. <https://doi.org/10.1557/mrs.2019.206>, Fig. 1f). In addition, the negative capacitance in OTS devices only seems to occur *below* the threshold voltage and seems to be related to the large current flowing through the device (see e.g. <https://doi.org/10.1088/0022-3727/16/5/021>). Related to this, I would like to quote from the paper "*Negative Capacitance Effect in Semiconductor Devices*":

"The incremental charge method of capacitance calculation is absolutely inapplicable in the case of large conduction current in the device, which is often the case when the capacitance is negative." (Ershov et al. IEEE Trans. Electron Devices 45, 2196 – 2206 (1998)).

From this we can conclude two things: 1) the negative capacitance effect reported by Vogel and Walsh in 1969 is not caused by a structural phase transition but is related to significant charge transport through the amorphous material, which invalidates the classical interpretation as a truly capacitive effect. 2) Our results show a genuine negative capacitance effect since charge transport through the device is negligible and the response is thus purely capacitive. Therefore, we insist on the novelty of our results.

R3: The point of confusion is that the phenomenon discussed by the authors and called them the "negative capacitance" is, strictly speaking, not the one. According to the textbook definition, the capacitance is given by the relation $C=V/Q$ where Q is the total charge at the electrodes. But what the authors observed is the local negative slope in the $V(Q)$ dependence, $dV/dQ<0$, occurring well far from the $Q=0$ point. This quantity is conventionally called the differential negative capacitance and appears in a rich multitude of manifestations, for example, in the mentioned above publication by Vogel R and Walsh PJ.

Response: We thank the reviewer for raising this important point. However, the textbook definition of capacitance $C = Q/V$ is only valid for strictly linear dielectric materials or vacuum in between the electrodes. Any non-linearity in the permittivity of a capacitor dielectric $\epsilon = dD/dE$, where D is the electric displacement field and E the electric field, automatically means that $C = Q/V$ cannot be used anymore (recall that $dQ = dD$ and $dV = t*dE$, where t is the distance between the electrodes). Therefore, the differential capacitance definition $C = dQ/dV$ is the more general definition since it can be applied for any linear or non-linear material. In the linear case, it simplifies to $C = dQ/dV = Q/V$. Importantly, a strictly linear negative capacitance as $C = Q/V < 0$ cannot occur in any passive dielectric medium, since it would be unstable for any operating point. Therefore, the differential negative capacitance region in any (anti)ferroelectric must always be bounded by regions of positive differential capacitance. For these reasons, when we say, “negative capacitance” we necessarily mean “differential negative capacitance” as the reviewer correctly noted. To emphasize this, we added the following to the manuscript on page 2:

“According to theory, the non-linear permittivity and thus capacitance of a material is proportional to $(d^2G/dP^2)^{-1}$, which means that at the antiferroelectric transition the capacitance of the material would become negative if stabilized in a larger system [17] [18]. Note that we always mean ‘negative differential capacitance’ when we write ‘negative capacitance’ in this context.”

R3: On the technical side, there are several prime importance flaws and insufficiently grounded statements.

1) The authors use the Kittel model of the ferroelectric-antiferroelectric transition to describe the experimentally observed hysteresis. However, even a brief comparison of the theoretical hysteresis plot (Fig.1) with the experimental observations (Fig.2) shows that these are entirely different phenomena.

Response: There seem to be a few misconceptions here. First, the Kittel model in Fig. 1 presents an idealized and simplified model of an antiferroelectric single crystal, which naturally will look different than a randomly oriented polycrystalline antiferroelectric film of 10 nm thickness, such as ours. Secondly, one of the major points of our manuscript is that the material we study (ZrO_2) does indeed not conform to the microscopic interpretation of the original Kittel model and other “classical” antiferroelectrics. We explicitly mention this in the abstract:

“Long-range polar or anti-polar order of such permanent dipoles gives rise to ferroelectricity or antiferroelectricity, respectively. However, the recently discovered antiferroelectrics of fluorite structure (HfO_2 and ZrO_2) are different: A non-polar phase transforms into a polar phase by spontaneous inversion symmetry breaking upon the application of an electric field.”

and in the main text:

“In contrast, the newly discovered HfO_2 and ZrO_2 based antiferroelectrics of fluorite structure transcend the classical definition of antiferroelectricity [11] [12], since their ground state does not exhibit anti-polar order but is microscopically non-polar [25].”

The reason why we still decided to use the Kittel model in Fig. 1 is twofold: 1) Historically, it was the first theory of antiferroelectricity and was published before any experimental evidence was

available that these materials even existed. We wanted to show that this original theory from 1951 already qualitatively predicted antiferroelectric negative capacitance, which no one had pointed out or investigated so far. 2) The Kittel model is a very well-known and illustrative model that explains the general phenomenology of the non-polar to polar antiferroelectric transition. Since there is currently no established phenomenological theory for antiferroelectricity in the new fluorite oxides like ZrO_2 , we decided to use the well-known Kittel model to motivate the qualitative prediction of antiferroelectric negative capacitance from a non-polar to polar phase transition, but not to quantitatively explain our results. We will discuss the model chosen for the NCFET simulations further below.

To emphasize this, we adjusted the text on page 4:

“Therefore, the original Kittel-model in Fig. 1a might not give a precise microscopic picture of antiferroelectricity in fluorite structure oxides and thus cannot explain the results quantitatively. However, the qualitative prediction of a negative capacitance from the non-polar to polar phase transition still applies.”

R3: Neither a single trace of the first-order phase transition nor an expected high-field saturation of the curve is observed experimentally.

Response: As mentioned before, the antiferroelectric P-V hysteresis in Fig. 2a is not expected to look the same as the theoretical curve for a single-crystal in Fig. 1a, because of the local variation of nanoscale grain sizes and orientations across the capacitor area. Such variations in the microstructure of polycrystalline thin films leads to local variations in the critical field and temperature, thus resulting in a broadening of the non-polar to polar phase transition (see e.g. <https://doi.org/10.1016/j.nanoen.2015.10.005>). Furthermore, the finite Thomas-Fermi screening length in the metal electrodes leads to a more “tilted” P-V hysteresis in very thin films (see e.g. the black curves in Fig. 1b and 2 in <https://doi.org/10.1063/1.2408650>). For these reasons, the slope dP/dV is less steep at the field-induced transition in our ZrO_2 films. The high-field saturation region cannot be seen in Fig. 2, because we did not want to apply more than 4 MV/cm to the device, which is already quite close to the typical breakdown field strength for 10 nm ZrO_2 and in accordance with previous reports (see e.g. <https://doi.org/10.1021/nl302049k>). However, for the 5 nm thick film shown in Fig. S2a, the expected high-field saturation can be observed since this thinner film did not break down for electric fields of up to 6 MV/cm.

R3: The experimentally observed “forbidden” branch, shown in Fig. 3d, starts at the point where the P(E) linearity loss occurs, which corresponds to the point M of the theoretical plot, rather than to the instability point B.

Response: As we have explained above, the quantitative difference between Fig. 1a and the measured data is related to the polycrystalline thin film nature of the 10 nm ZrO_2 layer in contrast to the single-crystal assumption of the Kittel model. However, out of curiosity we have tried to fit the Kittel-model with our experimental data, which is shown in Figure R5:

Figure R5. Comparison between experimental data of standard hysteresis measurement of TiN/ZrO₂/TiN (black lines), the pulsed voltage measurement of TiN/ZrO₂/Al₂O₃/HfO₂/TiN (red symbols) and the theoretical fit of the Kittel model (blue lines).

As can be seen from Fig. R5, despite the polycrystalline structure of our 10 nm ZrO₂ thin films, the single-crystal Kittel model of antiferroelectricity fits surprisingly well to the experimental data, including the negative capacitance regions. However, we decided not to include this figure in the manuscript or Supporting Information, since it might give the impression that the Kittel-model is suitable to describe the antiferroelectric ground state in ZrO₂, which is not the case as we have explained in response to one of the previous comments.

R3: Moreover, the “forbidden branch” does not join the high-field (paraelectric) branch, but goes somewhere else (Fig. S6c).

Response: A similar observation was also made by the second reviewer, which is why we will give a similar response here. The reason why we typically cannot measure the second high-field positive capacitance branch is the following: The large polarization in the antiferroelectric layer ($P > 15 \mu\text{C}/\text{cm}^2$) at these high external voltages, creates an additional electric field $E_{d,pol}$ across the dielectric layers, which has the same direction as the externally applied field $E_{d,ext}$, resulting in hard dielectric breakdown. Considering an effective dielectric of thickness t_d and permittivity ϵ_d (equivalent to the Al₂O₃/HfO₂ series stack), the total electric field in this dielectric is given by

$$E_d = E_{d,pol} + E_{d,ext} = \frac{P + \frac{\epsilon_0 \epsilon_{AFE}}{t_{AFE}} V_{ext}}{\epsilon_0 \left(\epsilon_d + \epsilon_{AFE} \frac{t_d}{t_{AFE}} \right)}$$

When we put our experimental parameters into this equation, we can estimate that $E_d > 10$ MV/cm for the highest applied voltages, which leads to hard dielectric breakdown of the capacitor. Similar behavior has been observed for ferroelectric/dielectric heterostructures, where often the second positive capacitance branch is not observed due to hard dielectric breakdown (see e.g. <https://doi.org/10.1038/s41586-018-0854-z>). However, on some of our fabricated

capacitors, which had a slightly better than average breakdown field strength, we were able to observe the second positive capacitance branch just before hard breakdown occurred (see Fig. R6). We added these data to the Supplementary information (Fig. S9) to confirm that we can indeed observe the second positive branch at high voltage if the sample has a better than average breakdown field strength. Regarding the few data points outside the P-V loop in Fig. S6c, these might be expected due to the broad phase transition caused by the polycrystalline nature of the sample. However, we expect that these would also join the high-field positive branch as in Fig. R6, if we could measure them at higher applied voltages (which was not possible due to dielectric breakdown).

Figure R6. Measurement of the second positive capacitance branch for high applied voltages. The measurement of the high polarization regions is limited by hard dielectric breakdown.

We added the following discussion to the manuscript on page 7:

“However, the second positive capacitance branch expected at higher fields is often not observed in experiments, since hard dielectric breakdown of the dielectric layers occurs at these high electric fields. Since dielectric breakdown is a statistical process, we were able to observe the second positive capacitance branch in some samples as shown in the Supplementary Information Fig. S9.”

R3: To justify the theoretical approach, the authors should present a theoretical fit of the experimental curve, based, in particular, on the given in SI S2 fitting coefficients, to justify the used parameters, and convincingly explain the apparent discrepancy between the data and the theory.

Response: We thank the reviewer for this suggestion. As we have explained before, we do not believe that the Kittel model is the correct quantitative model for antiferroelectricity in fluorite structure oxides due to the absence of an anti-polar order in the ground state. Therefore, we used a relatively simple Landau-based approach for the fitting of the P- E_a curves, see Fig. R7.

Figure R7. Landau fit of one of the negative capacitance regions for AFE NCFET simulations.

Since, in an NCFET device, only the direct vicinity of one of the negative capacitance regions will be used by shifting the P-V loop through gate metal work function engineering, we only fitted the right negative capacitance region with a simple 3rd order Landau polynomial as described in the Supplementary Section S2. As can be seen in Fig. R7, this simple model fits very well to the experimental data in the region of interest (the other regions are not accessible during NCFET operation, where the charge in the channel only changes by around $\sim 2 \mu\text{C}/\text{cm}^2$ between the “on”- and “off”-state). We added this figure to the Supplementary Information as Fig. S11. For NCFET applications, having a very thin gate oxide layer is beneficial, which is why we chose 1.8 nm for this simulation. Experimentally, it is known that the crystal structure of antiferroelectric ZrO_2 becomes more and more distorted with decreasing thickness (i.e., the tetragonality or c/a ratio increases). It has been found that this leads to an increase in the distance between the critical fields for the tetragonal-to-orthorhombic and the orthorhombic-to-tetragonal phase transitions (see <https://doi.org/10.1002/aelm.202100485>). To accommodate for this experimentally observed increase of tetragonality, we increased the “width” of the fitted S-curve in Fig. R7 from $E_H = 1.8 \text{ MV/cm}$ to 3 MV/cm for the simulation results presented in Fig. S13. The fitted value of $P_o^{\text{AF}} = 15 \mu\text{C}/\text{cm}^2$ was left unchanged.

To better explain the rationale behind these coefficients, we added the following to the Supplementary Information Section S2:

“For our simulations, we first fitted $P_o^{\text{AF}} = 15 \mu\text{C}/\text{cm}^2$ and $E_H = 1.8 \text{ MV/cm}$ to the experimental data for 10 nm ZrO_2 as shown in Fig. S11. However, recently, it was found that the distance between the critical fields of the tetragonal-to-orthorhombic and the orthorhombic-to-tetragonal phase transitions increases with decreasing ZrO_2 thickness, due to an increase in the tetragonality of the unit cell.[5] To accommodate for this increased tetragonality expected for a 1.8 nm thick ZrO_2 film in an NCFET, we increased E_H to 3 MV/cm for the simulation.”

R3: 2) The experimental foundation of the work, in particular the characterization of the hysteretic phenomena, needs further clarification. The absence of the saturation point and the profound hysteresis raises the question about its thermodynamic origin, in particular, whether the low-field and high-field branches are locally reversible? Authors should use the additional

protocols of the field sweep with different sweep-inversion points to establish that both branches are stable and unique and that no other history-dependent hysteresis paths arise and interfere in the P-E space. Only in this case, the attribution of the hysteresis branches to particular phases and the thermodynamic consideration of the NC would make sense.

Response: We thank the reviewer for mentioning this important point. As we have discussed and shown in response to the previous comments, we can indeed measure the saturation point if the breakdown field strength of the sample is high enough. However, another important point when discussing hysteresis with respect to our experiments is the subtle effect of leakage currents and charge injection.

At the highest applied voltages, the electric fields in the dielectric layers can become close to the breakdown field strength due to the reasons we mentioned before. These high electric fields can enable Fowler-Nordheim tunneling of electrons into the conduction band of the dielectric layers. Some of these tunneling electrons might then be trapped in deep energetic states at the antiferroelectric/dielectric interface. In our pulsed voltage measurements (e.g. Fig. 3), any leakage contribution will manifest as a residual charge Q_{res} (see Fig. 3c), which is the difference between charging and discharging curves of the capacitor.

In the case that some fraction of Q_{res} is trapped at the antiferroelectric/dielectric interface, let's call this charge Q_{trap} , this would change the electrostatic boundary condition at the antiferroelectric/dielectric interface, since $D_{DE} = D_{AFE} + Q_{trap}/A$, where D_{DE} and D_{AFE} are the electric displacement fields in the dielectric and antiferroelectric, respectively and A is the capacitor area. Thus, the charge density on the capacitor electrodes is then given by $Q/A = D_{DE} \approx P + Q_{trap}/A$. Therefore, Q_{trap} will shift our extracted P- E_a curve, where we assumed that $Q/A \approx P$ (since we cannot exactly determine Q_{trap} from these experiments). This effect can thus lead to an apparent hysteresis in the P- E_a curve when there is a significant charge Q_{res} . However, this hysteresis does not come from the antiferroelectric layer, but from the parasitic effects of leakage and subsequent charge trapping. For this reason, we must be careful when interpreting hysteresis in the P- E_a curves from the pulsed measurements.

Fig. R8 shows the experimental data for using different voltage sweep directions. We start with voltage pulses from 0 V in small steps until we reach around 11 V as the maximum pulse voltage from which we decrease the voltage pulse height in small steps until we reach 0 V again. As we can see, we do trace back the negative capacitance branch with descending voltage pulse levels. The small hysteresis can be explained by a small amount of charge injection during the application of the highest voltage pulses (Q_{res} is not completely zero), which slightly shifts the original P- E_a curve. For negative applied pulses, the situation is different: significant hysteresis is observed, which is related to the large Q_{res} measured for this polarity (see Fig. 3c), leading to substantial charge trapping. This strongly asymmetric hysteresis and the correlation with Q_{res} shows that this hysteresis is a parasitic effect caused by charge trapping which is not related to the antiferroelectric phase transition itself (which should be identical for positive and negative voltages).

Figure R8. Pulsed voltage hysteresis measurement starting from 0 V to 11 V back to 0 V to -11 V and back to 0 V. Very small hysteresis is observed for positive applied voltages, since the leakage and charge trapping is small (Q_{res} is low). For negative applied voltages the hysteresis is large due to leakage and subsequent charge trapping (Q_{res} is high).

We added this figure to the Supplementary Information as Fig. S12. Furthermore, we added the following discussion on page 7 in the manuscript:

“Furthermore, in Fig. S12 we incrementally changed the voltage pulse amplitude from 0 V \rightarrow 11 V \rightarrow -11 V \rightarrow 0 V to investigate the reversibility of the P - E_a curve. An asymmetric hysteresis emerges which seems to correlate with the observation of significant Q_{res} for negative voltages in Fig. 3c. This suggests that the hysteresis is not related to the non-polar to polar phase transition itself, but that it is caused by leakage and subsequent charge trapping of a fraction of Q_{res} at the antiferroelectric/dielectric interface, which leads to a shift of the apparent P - E_a curve. For positive voltages, where Q_{res} is low, negative capacitance is observed in both forward and backwards sweep directions in Fig. S12.”

R3: 3) The possible role of domains of different types that easily arise in films, especially in the AFE phase is not clarified in the article. Even the possibility of the presence of domains of different types is not verified experimentally. To formulate the description of domains resting on the Kittel theory, the authors should experimentally demonstrate that their hysteresis curve is due to a single-phase transformation rather than being caused by the collective domain switching. The microscopy-scale pictures of the domain pattern and the field dependencies of domain configurations should be presented.

Response: We thank the reviewer for bringing up this important point. As we have mentioned in response to some of the previous comments of the reviewer, antiferroelectric ZrO_2 is different from other “classical” antiferroelectrics in that the macroscopic non-polar state is caused by a microscopic non-polar crystal structure. Each tetragonal $P4_2/nmc$ unit cell in the ground state is centrosymmetric which is consistent with our experimental GIXRD, STEM and nanobeam electron diffraction data. Since the tetragonal phase is centrosymmetric, there are no domains to be observed in the antiferroelectric ground state. For this reason, we did not use the Kittel model for the device simulation in Section S2.

As mentioned previously, the field-induced non-polar to polar phase transition is still unlikely to happen in the whole film at the same electric field, due to the polycrystalline morphology of the film, which will lead to a spatial variation of the transition field across the capacitor area. Observing this local transition directly e.g., in high-resolution TEM with an applied electric field

has been the goal of many leading groups (including ours) for many years, with limited success. Therefore, the technical challenges of such experiments and the necessary time investment should not be underestimated. Thus, such experiments are beyond the scope of the current manuscript, which focuses on the observation of negative capacitance in antiferroelectric ZrO_2 and not on the direct microscopic observation of the non-polar to polar phase transition, which in and of itself would be its own major study. Only very recently (after the initial submission of this manuscript), a first report of a successful direct measurement of such a phase transition has been reported in ferroelectric $\text{Hf}_{0.5}\text{Zr}_{0.5}\text{O}_2$ (Y. Zheng et al., "In-situ atomic visualization of structural transformation in $\text{Hf}_{0.5}\text{Zr}_{0.5}\text{O}_2$ ferroelectric thin film: from nonpolar tetragonal phase to polar orthorhombic phase," 2021 Symposium on VLSI Technology, 2021, pp. 1-2.).

To make this topic clearer to the reader, we added the following discussion and the new reference to the manuscript on page 4:

"Since the structural data suggests that our ZrO_2 films are fully tetragonal without applied voltage, the measured double hysteresis loops can only be explained by a field-induced structural transformation into a polar ferroelectric phase. While we cannot directly determine the symmetry of this polar phase at high electric field, there is substantial evidence in literature which suggests that it is of orthorhombic $Pca2_1$ symmetry [26]. For example, a temperature-dependent phase transition from the non-polar $P4_2/nmc$ phase to the polar $Pca2_1$ phase in similar ZrO_2 thin films has been experimentally observed with in situ with high-temperature X-ray diffraction [31]. Furthermore, ZrO_2 can be stabilized in the $Pca2_1$ phase even at room temperature under certain processing conditions [32]. Lastly, the field-induced transition from the $P4_2/nmc$ phase to the $Pca2_1$ phase has been directly observed in $\text{Hf}_{0.5}\text{Zr}_{0.5}\text{O}_2$ thin films [33]. From these experimental data and previous first principles calculations [25] [29], it seems reasonable to conclude that the polar phase observed at high electric field in Fig. 2a is of $Pca2_1$ symmetry."

R3: 4) All the above points are critical for achieving an unambiguous interpretation of the assumed observation of the differential NC in the "forbidden region". To summarize: (i) The experimental data, presented in Fig. 3 are insufficient to judge whether the supposed NC "forbidden" branch is reversible. To clarify the reversibility, the authors should use the reverse protocols of the gradual charge removing from the highest charge-carrying state and demonstrate that the "forbidden" hysteretic branch returns back along the same thermodynamic path. Furthermore, to confirm the S-curve description, the authors should reveal experimentally, how their "forbidden branch" joins the high-field stable hysteretic branch. (ii) To build on their statement that the (differential) NC effect is caused by the structural phase transition, the authors should either experimentally demonstrate that the system is in a single-domain state or, if this is not the case, to clarify the impact of a many-domain structure on the assumed phenomena manifestations.

Response: We thank the reviewer for this constructive feedback, which has enabled us to substantially improve our manuscript. As the reviewer suggested we have carried out additional experiments showing the reversibility of the NC region (Fig. S12), where the asymmetric hysteresis can be explained by parasitic charge trapping effects (otherwise, it would be symmetric). Furthermore, we have shown that the negative capacitance region indeed joins the high-field positive capacitance branch in samples that have a higher-than-average breakdown field strength (Fig. S9). In the other samples, the very high electric field across the dielectric

layers which leads to hard dielectric breakdown prevents the observation of this high-field branch.

Regarding the second point, we have experimentally shown with various methods (GIXRD, STEM and nanobeam electron diffraction) that our ZrO₂ films are in a completely non-polar (tetragonal P4₂/nmc) ground state at zero volts, which means that our electrical characteristics (Fig. 2a) can only be interpreted as a structural phase transition to a polar phase. Recent experimental and theoretical literature results suggest that this polar phase has Pca2₁ symmetry. Due to the substantial technical challenges of directly imaging the electric field induced phase transition *in situ* (which we have tried for a long time), we feel that such experiments are beyond the scope of the present study. As we have emphasized in the revised manuscript, the experimental evidence presented (fully non-polar tetragonal ground state and antiferroelectric hysteresis) cannot be explained without a non-polar to polar structural phase transition of the ZrO₂ film. Understanding the detailed local variations and dynamical behavior of this structural transition related to the nanoscale polycrystalline structure of ZrO₂ should be the goal of future studies. However, since it is known that these nanoscale (anti)ferroelectric grains in fluorite structure oxides act independently of each other (the switching behavior of each grain does not depend on the other grains), the macroscopic response of a capacitor represents the behavior that is expected for an “average” grain taken from the film.

R3: Minor comments:

5) Since the differential NC effect takes place at some finite (working) voltage it will be better to indicate the real values of the voltage in Figs. S9, rather than the relative ones. It is also advisable to indicate in Supplementary S2, the value of such working voltage for the modeled transistor and whether such values are operational in modern nanoelectronics. A broad readership would benefit from having access to direct data.

Response: Thank you for this comment. Indeed, we have considered the fact that the antiferroelectric negative capacitance region only appears at a finite voltage. However, the NCFET device can be designed to compensate for this voltage offset by applying work function engineering as has investigated theoretically before (see <https://doi.org/10.1109/LED.2017.2733382>). Furthermore, in the context of antiferroelectric memory devices, similar shifting of one of the hysteresis-loops by work function engineering has been experimentally demonstrated (see <https://doi.org/10.1002/adfm.201603182>). We used the same approach for our antiferroelectric NCFET simulations in Fig. S13. Since the thickness of the antiferroelectric layer in the simulation is 1.8 nm, we need about 400 mV of work function shift to obtain an internal bias field of ~2.2 MV/cm, which is enough to center one of the negative capacitance regions around zero gate voltage (see Fig. 3d). Such voltage shifts obtained by gate metal work function engineering are experimentally feasible and have been demonstrated in the past (see e.g. DOI: 10.1109/LED.2003.809528). Therefore, the operating voltages reported in Fig. S13 are indeed the real voltages under which such a device would operate. These voltage levels of 0.8 V and below are practical for modern nanoelectronic devices and circuits.

R3: 6) Many key statements of the Manuscript are formulated in a vague and fogging manner.

For example:

- “An applied electric field of around 2-3 MV/cm can then transform the non-polar tetragonal phase into the polar orthorhombic Pca21 phase, which is known to be ferroelectric [26]. While obtaining definitive experimental proof of such a field-induced first-order phase transition has proved difficult so far [27], this mechanism is consistent with first-principles calculations [24] [28] as well as composition- and temperature-dependent experimental results [25]. While the microscopic switching pathway between the P42/nmc and Pca21 phases is still unclear, it has been suggested to include an intermediate phase of orthorhombic Pmn21 symmetry for Hf_{0.5}Zr_{0.5}O₂ [29].”

Authors should indicate clearly, whether the ferroelectric nature of the high-field phase is proven or this is just their hypothesis not actually confirmed by the experiment. Reference [26] is misleading. The statement that “the polar orthorhombic Pca21 phase, ... is known to be ferroelectric” is not the achievement of Ref. [26] but is the subject of the textbook knowledge. The way the reference is given provokes the feeling that Ref. [26] proves that the high-field phase observed in the present manuscript is ferroelectric as well.

Response: We thank the reviewer for this important feedback. Our goal is for the manuscript to be as clear and unambiguous as possible. Therefore, we have rewritten the text in question:

“The current understanding of the origin of antiferroelectricity in ZrO₂ is that the non-polar tetragonal P4₂/nmc phase undergoes a first-order structural phase transition into the polar orthorhombic Pca2₁ phase by application an electric field of around 2-3 MV/cm [25] [26]. The polar orthorhombic Pca2₁ phase has been shown to be responsible for the ferroelectric behavior observed in HfO₂ based thin films [27].”

R3: - “Supplementary Fig. S1: Grazing-incidence X-ray diffraction measurement of ZrO₂. Diffraction patterns and their indices for tetragonal ZrO₂ as well as TiN and Al phases are marked in the figure. For comparison, the position of the diffraction peaks of orthorhombic ZrO₂ and monoclinic ZrO₂ are indicated at the bottom. The Bragg peaks for ZrO₂ match well with those of the tetragonal structure. No resemblance of the diffraction patterns of our samples with the orthorhombic and monoclinic patterns is observed in these samples indicating that the crystalline phase in these samples are predominantly tetragonal with negligible fractions (below the detection limit of our XRD set-up) of orthorhombic and monoclinic phases”.

The difference in patterns and the absence of the orthorhombic or monoclinic phases are different statements. The authors should indicate clearly, whether the fractions of orthorhombic and monoclinic phases were observed or not.

Response: Thank you for pointing this out. We agree with the reviewer that this statement was not clear enough and changed the caption of Supplementary Fig. S1 accordingly to:

“No fractions of the orthorhombic and monoclinic phase were observed in our samples based on X-ray diffraction data, which is consistent with scanning transmission electron microscopy and nanobeam electron diffraction results (see Fig. 2b and Fig. S3).”

REVIEWER COMMENTS

Reviewer #1 (Remarks to the Author):

I think most of my concerns have been addressed and the manuscript has been revised accordingly. Especially, the rationale for claiming that this NC effect comes from a nonpolar to polar antiferroelectric-like transition has been made clear.

However, I have one concern regarding the role of domains (or lack thereof) which is probably similar to one of reviewer 3's comments. Although the authors have checked that ZrO₂ in the as-prepared capacitor is in the nonpolar tetragonal phase, they have not checked whether that is the case after voltage cycling.

So, as far as I understand, they cannot rule out the possibility of domain motion in the polar phase as the source of the antiferroelectric-like P-E behavior. Could this be resolved by structural analysis of the film at 0 V after voltage application?

Reviewer #2 (Remarks to the Author):

The authors has addressed the questions we proposed for consideration.

Response to Reviewers

Reviewer #1:

R1: I think most of my concerns have been addressed and the manuscript has been revised accordingly. Especially, the rationale for claiming that this NC effect comes from a nonpolar to polar antiferroelectric-like transition has been made clear.

However, I have one concern regarding the role of domains (or lack thereof) which is probably similar to one of reviewer 3's comments. Although the authors have checked that ZrO₂ in the as-prepared capacitor is in the nonpolar tetragonal phase, they have not checked whether that is the case after voltage cycling.

So, as far as I understand, they cannot rule out the possibility of domain motion in the polar phase as the source of the antiferroelectric-like P-E behavior. Could this be resolved by structural analysis of the film at 0 V after voltage application?

Response: We thank the reviewer for bringing up this important point. We understand the concern of the reviewer that the ZrO₂ film might be in a multidomain state after the voltage application at 0V. Indeed, this is something we were considering in the initial phase of our experiments. For this reason, we have previously carried out high-resolution TEM experiments with *in situ* voltage biasing, which have already been published (see DOI: 10.1109/VLSITechnology18217.2020.9265091; Ref. 28 in the manuscript). The TiN/ZrO₂/TiN sample investigated in Ref. 28 was fabricated in the same way and at a similar time as the TiN/ZrO₂/TiN sample investigated in the present study (Fig. 2a). Therefore, the results of ref. 28 are directly relevant for the interpretation of the results in the present manuscript. What we found in Ref. 28 was the following:

Initially, at 0V, the ZrO₂ film was found to be in the tetragonal P4₂/nmc phase based on nano-beam electron diffraction (NBED) as well as HRTEM. This is fully consistent with the GIXRD, NBED and STEM results reported on TiN/HfO₂/Al₂O₃/ZrO₂/TiN samples in the present manuscript. With *in situ* biasing up to 4V in HRTEM, we observed a change in the crystal structure of the ZrO₂ layer, which returned to the initial structure after the bias was removed as can be seen in Fig. R1:

Figure R1. HRTEM of the ZrO₂ layer before (left), during (middle) and after (right) *in-situ* biasing experiments at 4V. Yellow circles for 0V correspond to the (11 $\bar{2}$) projection of Zr atoms in the tetragonal P4₂/nmc phase. The local crystal structure before and after *in-situ* biasing was identical. Figure taken from Ref. 28, Fig. 6.

This direct evidence of a reversible phase transformation was also observed after each subsequent *in situ* biasing to 4V (see also ref. 28, Fig. 9). Furthermore, even after 10^7 voltage cycles, the macroscopic P-E loop was virtually unchanged (see ref. 28, Fig. 4), consistent with no change in the non-polar ground state with cycling. These measurement results provide strong evidence that the crystal structure of the antiferroelectric ZrO_2 layer after the field-induced phase transition is the same as the initial one, which was consistently found to be the tetragonal $P4_2/nmc$ phase in both $TiN/ZrO_2/TiN$ and $TiN/HfO_2/Al_2O_3/ZrO_2/TiN$ samples. Since the ZrO_2 layers in Ref. 28 and in the present manuscript were fabricated in the exact same way (same ALD tool, growth parameters, annealing conditions etc.) and at a similar time, we have no reason to believe that the $TiN/HfO_2/Al_2O_3/ZrO_2/TiN$ stacks would behave differently in HRTEM with *in-situ* biasing under similar electric fields. Since such *in situ* biasing experiments are very difficult and time consuming (taking many months) to perform, repeating them on $TiN/HfO_2/Al_2O_3/ZrO_2/TiN$ samples does not seem reasonable to us. Especially, since the ZrO_2 layer seems to have the same initial structure as the sample in Ref. 28, which we expected due to their identical fabrication procedure.

To make this point clear to the reader, we have added the following sentences to the manuscript on page 4:

“Furthermore, using high-resolution transmission electron microscopy (HRTEM) with in situ voltage biasing, it was directly shown that antiferroelectric ZrO_2 always returns to its initial non-polar $P4_2/nmc$ structure after the applied voltage is removed [28]. Since the ZrO_2 film investigated in Ref. [28] was fabricated in the exact same way as the ones shown here, it is reasonable to assume that they also return to the initial non-polar $P4_2/nmc$ phase after each field-induced phase transition.”

We also added the following on page 5:

“As mentioned before, previous in situ HRTEM experiments showed that these ZrO_2 layers always return to their initial non-polar ground state after the applied voltage is removed [28].”

We hope that these explanations and additions to the manuscript will address the last remaining concerns and convince the reviewer that our revised manuscript is now suitable for publication in Nature Communications.

REVIEWERS' COMMENTS

Reviewer #1 (Remarks to the Author):

Although there may be some doubt as to assuming that the ZrO₂ layers in TiN/ZrO₂/TiN and TiN/HfO₂/Al₂O₃/ZrO₂/TiN should behave similarly, I can understand that further confirmation would be rather costly and time-consuming.

Thus, I think that the authors have addressed the concerns put forth in the previous reviewer report to an extent that is reasonably achievable.

Response to Reviewers

Reviewer #1:

R1: Although there may be some doubt as to assuming that the ZrO_2 layers in $\text{TiN/ZrO}_2/\text{TiN}$ and $\text{TiN/HfO}_2/\text{Al}_2\text{O}_3/\text{ZrO}_2/\text{TiN}$ should behave similarly, I can understand that further confirmation would be rather costly and time-consuming.

Thus, I think that the authors have addressed the concerns put forth in the previous reviewer report to an extent that is reasonably achievable.

Response: We want to thank the reviewer for the positive feedback and appreciate the time that was taken to review our manuscript. The constructive criticism has led to a significant improvement of the overall quality of the manuscript.